# miR200-regulated CXCL12β promotes fibroblast heterogeneity and immunosuppression in ovarian cancers

Anne-Marie Givel[1,2], Yann Kieffer[1,2], Alix Scholer-Dahirel[1,2], Philemon Sirven[3], Melissa Cardon[1,2], Floriane Pelon[1,2], Ilaria Magagna[1,2], Géraldine Gentric[1,2], Ana Costa[1,2], Claire Bonneau[1,2], Virginie Mieulet[1,2], Anne Vincent-Salomon[4] & Fatima Mechta-Grigoriou [1,2]

High-grade serous ovarian cancers (HGSOC) have been subdivided into molecular subtypes. The mesenchymal HGSOC subgroup, defined by stromal-related gene signatures, is invariably associated with poor patient survival. We demonstrate that stroma exerts a key function in mesenchymal HGSOC. We highlight stromal heterogeneity in HGSOC by identifying four subsets of carcinoma-associated fibroblasts (CAF-S1-4). Mesenchymal HGSOC show high content in CAF-S1 fibroblasts, which exhibit immunosuppressive functions by increasing attraction, survival, and differentiation of CD25[+]FOXP3[+] T lymphocytes. The beta isoform of the CXCL12 chemokine (CXCL12β) specifically accumulates in the immunosuppressive CAF-S1 subset through a miR-141/200a dependent-mechanism. Moreover, CXCL12β expression in CAF-S1 cells plays a crucial role in CAF-S1 immunosuppressive activity and is a reliable prognosis factor in HGSOC, in contrast to CXCL12α. Thus, our data highlight the differential regulation of the CXCL12α and CXCL12β isoforms in HGSOC, and reveal a CXCL12β-associated stromal heterogeneity and immunosuppressive environment in mesenchymal HGSOC.

[1] Institut Curie, Stress and Cancer Laboratory, Equipe labelisée Ligue Nationale Contre le Cancer, PSL Research University, 26, rue d'Ulm, 75005 Paris, France. [2] U830, Inserm, 75005 Paris, France. [3] Institut Curie, Integrative Biology of Human Dendritic Cells and T Cells Laboratory, PSL Research University, U932, Inserm, 26, rue d'Ulm, 75005 Paris, France. [4] Department of Pathology, Institut Curie Hospital Group, 26, rue d'Ulm, 75248 Paris, France. Correspondence and requests for materials should be addressed to F.M-G. (email: fatima.mechta-grigoriou@curie.fr)

High-grade serous epithelial ovarian cancers (HGSOC), commonly treated by the combination of surgery and chemotherapy, remain one of the deadliest gynecologic malignancies. Despite an initial response to treatment, many patients relapse, become resistant, and ultimately die. To date, treatment strategy mainly relies on clinico-pathologic aspects, such as histological type, grade and stage without consideration of molecular phenotypes. HGSOC genomic and transcriptomic profiles have been helpful for characterizing HGSOC molecular features and improving patient stratification leading to new treatment strategies. HGSOC patients carrying *BRCA1/2* alterations have increased sensitivity to platinum salts and a longer survival than non-mutated patients, and are now eligible for anti-PARP therapies[1–5]. In addition to genomic characterization, several groups have defined distinct HGSOC molecular subtypes based on transcriptomic profiling[6–13]. In all studies, one molecular subgroup, referred to as "Fibrosis" or "Mesenchymal", has been systematically identified and is invariably associated with poor patient survival. Interestingly, one of the first mechanisms that differentiates the Fibrosis/Mesenchymal HGSOC from the other molecular subtypes depends on the miR-200 family of microRNA[7,13,14]. Still, patients suffering from HGSOC of the Fibrosis/Mesenchymal subtype invariably show poor prognosis and remain one of the major clinical challenges in ovarian tumorigenesis.

Transcriptomic signatures that identify the "Fibrosis/Mesenchymal" HGSOC tumors[6–13] include several genes involved in matrix remodeling and stromal components, suggesting a specific role of the stroma in this HGSOC molecular subtype. Carcinoma-associated fibroblasts (CAF) are one of the most abundant components of the tumor microenvironment and represent attractive targets for therapeutic intervention. Several studies have demonstrated that the proportion of CAF in ovarian cancers is associated with poor prognosis[15,16]. These cells contribute to tumor initiation, metastasis[17–20], and resistance to treatment[21]. However, CAF identification and molecular characterization remain poorly defined in HGSOC, and nothing is known about CAF features in the "Fibrosis/Mesenchymal" molecular subtype.

Our study highlights new biological properties of the mesenchymal HGSOC. We describe for the first time stromal heterogeneity in HGSOC by identifying four CAF subpopulations (CAF-S1−S4). Moreover, we show that accumulation of the CAF-S1 subset in mesenchymal HGSOC is associated with an immunosuppressive environment. While the role of the chemokine (C-X-C motif) ligand 12 (CXCL12) on HGSOC patient survival remains controversial and the impact of the different CXCL12 isoforms is still largely unknown[22–24], we highlight here that *CXCL12α* and *CXCL12β* isoforms accumulate differentially in the two subsets of activated fibroblasts identified (namely CAF-S1 and CAF-S4). Indeed, the *CXCL12β* isoform specifically accumulates in the CAF-S1 subpopulation, and not in the CAF-S4 subset. This differential accumulation results from a post-transcriptional mechanism, dependent of miR-200 family members, miR-141 and miR-200a. The expression of these two miRNA leads to the specific downregulation of the *CXCL12β* isoform in CAF-S4 fibroblasts and subsequently to its accumulation in CAF-S1 immunosuppressive fibroblasts. Regulation of *CXCL12* isoforms in CAF-S1 plays a key role in mesenchymal HGSOC. Indeed, the expression of *CXCL12β* by CAF-S1 fibroblasts is essential for T-cell attraction towards CAF-S1-enriched HGSOC. Once attracted, CAF-S1 fibroblasts enhance the survival, as well as the content in $CD25^+FOXP3^+$ T lymphocytes. This latter effect is independent of CXCL12, but mediated through B7H3, CD73, and IL6 that are highly expressed in CAF-S1 cells. Thus, our work highlights for the first-time stromal heterogeneity

in HGSOC and uncover the specific regulation and function of the *CXCL12β* isoform in defining stromal and immune features in mesenchymal HGSOC, one of the most deleterious subtypes of ovarian cancers.

## Results

**Mesenchymal HGSOC exhibit CAF heterogeneity**. Gene signatures defining HGSOC of the mesenchymal subtype are all composed of stromal genes[6–12]. We hypothesized that stroma could play an important role in the development of mesenchymal HGSOC. We first evaluated stromal quantity and cellular density in HGSOC. We observed that mesenchymal HGSOC exhibited higher stromal content than non-mesenchymal tumors (Fig. 1a, b). Moreover, stroma from mesenchymal HGSOC was compact and tight with high fibroblast cellularity (defined as "dense"), while non-mesenchymal tumors showed scattered and sprinkled stroma with low cellularity (defined as "loose") (Fig. 1a, c). We next aimed at performing deeper characterization of CAF in HGSOC. To do so, we performed multicolor flow-cytometry (fluorescence-activated cell sorting (FACS)) (Fig. 1) and immunohistochemistry (IHC) analyses (Fig. 2) (Table 1 for details on retrospective cohorts) using concomitantly six different markers, including FAP (fibroblast activation protein), CD29 (integrin-β1), SMA (smooth muscle α-actin), FSP1 (fibroblast-specific protein 1), PDGFRβ (platelet-derived growth factor receptor-β), and caveolin (see Methods and Supplementary Table 1 for antibody references). To our knowledge, no study analyzing simultaneously six fibroblast markers has ever been done in ovarian cancers. Among viable cells detected by FACS using fresh human HGSOC, we identified epithelial, immune, and endothelial cells using EPCAM, CD45, and CD31 markers, respectively (Fig. 1d, left). Fibroblasts were considered as being part of the $EPCAM^-CD45^-CD31^-$ cells and were further characterized with the above-mentioned fibroblast markers (Fig. 1d, right). Interestingly, we distinguished four different CAF sub-populations in HGSOC, according to CD29, FAP, FSP1, and SMA protein levels (Fig. 1d, right). These four CAF subsets were named CAF-S1 (red), CAF-S2 (yellow), CAF-S3 (green), and CAF-S4 (blue) (Fig. 1d, right). To confirm the existence of the four different CAF subsets in ovarian tumors, we used an unsupervised algorithm, named Cytospade[25], an open source platform for network analysis that organizes cells into hierarchies of related phenotypes. The trees constructed by applying Cytospade to FACS data enabled us to validate the presence of four CAF subsets in HGSOC (Fig. 1e). Two populations, CAF-S2 and CAF-S3, can be defined as "non-activated" CAF based on the lack of expression of SMA, while both CAF-S1 and CAF-S4 expressed SMA and can be considered as "activated" CAF or myofibroblasts (Fig. 1f). CAF-S1 fibroblasts expressed high levels of all markers tested, as opposed to CAF-S2 that were negative for all (Fig. 1f). CAF-S3 and CAF-S4 were positive for specific but distinct markers (Fig. 1f). Indeed, CAF-S4 did not express FAP but showed high levels of CD29 and SMA proteins, while CAF-S3 exhibited intermediate to low levels of CD29 and SMA markers, but high FSP1 protein levels. Of note, caveolin was not detected in CAF-S2 and showed low levels in the other three CAF subsets (Fig. 1f), indicating that this fibroblastic marker was not helpful for differentiating CAF subsets in HGSOC and will not be used further in our study. Thus, CAF subsets can be defined by the following profiles in HGSOC: CAF-S1: $CD29^{Med-Hi}$ $FAP^{Hi}$ $SMA^{Med-Hi}$ $FSP1^{Med-Hi}$ $PDGFRβ^{Med-Hi}$ $CAV1^{Low}$; CAF-S2: $CD29^{Low}$ $FAP^{Neg}$ $SMA^{Neg-Low}$ $FSP1^{Neg-Low}$ $PDGFRβ^{Neg-Low}$ $CAV1^{Neg}$; CAF-S3: $CD29^{Med}$ $FAP^{Low}$ $SMA^{Low}$ $FSP1^{Med-Hi}$ $PDGFRβ^{Med}$ $CAV1^{Neg-Low}$; CAF-S4: $CD29^{Hi}$ $FAP^{Low}$ $SMA^{Hi}$ $FSP1^{Hi}$ $PDGFRβ^{Med-Hi}$ $CAV1^{Neg-Low}$.

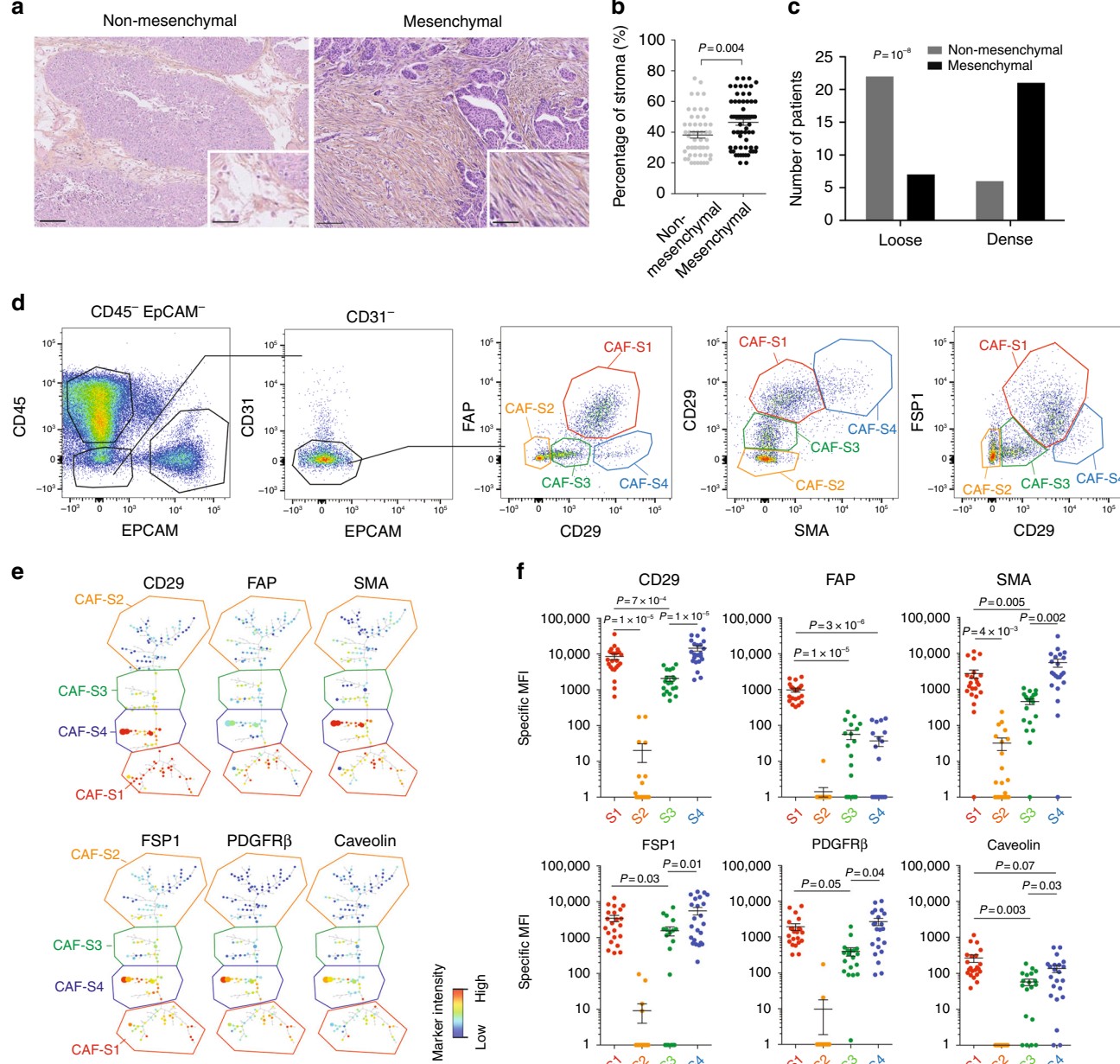

**Fig. 1** Identification of four CAF subsets in HGSOC. **a** Representative view of HES staining of non-mesenchymal and mesenchymal HGSOC sections (Institut Curie cohort). Scale bar, 100 µm (low magnification) and 40 µm (inset). **b** Scatter plot showing the percentage of stroma in HGSOC. N = 107. Data are shown as mean ± SEM. P values are from Mann-Whitney test. **c** Bar plot showing association of mesenchymal HGSOC with stromal features, defined by pathologists as loose (low stromal cellularity) or dense (high stromal cellularity). N = 56. P value is from Fisher's Exact Test. **d** Gating strategy to identify CAF subsets in HGSOC by FACS. Results from a representative HGSOC patient are shown. Cells isolated from freshly dissociated human HGSOC were first gated on DAPI−, EPCAM−, CD45−, CD31− cells, for excluding dead cells (DAPI+), epithelial cells (EPCAM+), hematopoietic cells (CD45+), and endothelial cells (CD31+). Selected cells were next examined using six fibroblast markers. Representative flow cytometry plots show gating strategies based on FAP, CD29, SMA, and FSP1 that allow the identification of four sub-populations of fibroblasts in HGSOC: CAF-S1 are CD29$^{Med}$ FAP$^{High}$ SMA$^{High}$ FSP1$^{High}$, CAF-S4 are CD29$^{High}$ FAP$^{Low}$ SMA$^{High}$ FSP1$^{Med}$, CAF-S3 are CD29$^{Low}$ FAP$^{Low}$ SMA$^{Low}$ FSP1$^{Med/High}$ and CAF-S2 are CD29$^{Low}$ FAP$^{Low}$ SMA$^{Low}$ FSP1$^{Low}$. **e** CytoSpade trees annotated with each marker expression in HGSOC analyzed by FACS. Colors show staining intensity for each marker. Size of the nodes is proportional to the number of cells showing similar staining for the markers analyzed. **f** Scatter plots showing specific mean fluorescent intensity (MFI) detected for each marker in each CAF sub-population. Each dot represents the specific median of fluorescent intensity of the cellular population by patient. N = 22. Data are shown as mean ± SEM. P values are from Student's t-test

The existence of the four CAF sub-populations was validated by IHC on serial sections of HGSOC samples (Fig. 2a), using five out of the six fibroblast markers listed above, except caveolin (see Supplementary Table 1 for list of antibody references and Supplementary Fig. 1a for isotype controls). We evaluated histological scores (H score) for each marker in the stromal compartment. We applied a decision tree algorithm (Fig. 2b) to determine the global CAF subset enrichment per tumor. In brief, this decision tree was based on marker intensity thresholds, first defined according to FACS data (intensities of each CAF marker in each CAF subset) from fresh HGSOC samples and next transposed to IHC values (see Methods, #Development of a

decision tree algorithm for prediction of CAF subset identity). HGSOC were mainly enriched in activated CAF-S1 and CAF-S4 subsets (Fig. 2c). Interestingly, mesenchymal HGSOC accumulated more CAF-S1 fibroblasts than non-mesenchymal tumors (Fig. 2d). In parallel, we developed a method combining R script

and Fiji plugins to stack IHC staining of serial HGSOC sections. This method provided maps, where each square corresponded on average to one cell. Histological scores of the five CAF markers in each square defined CAF subset identity at cellular level and allowed to visualize their geographic repartition within the tumor

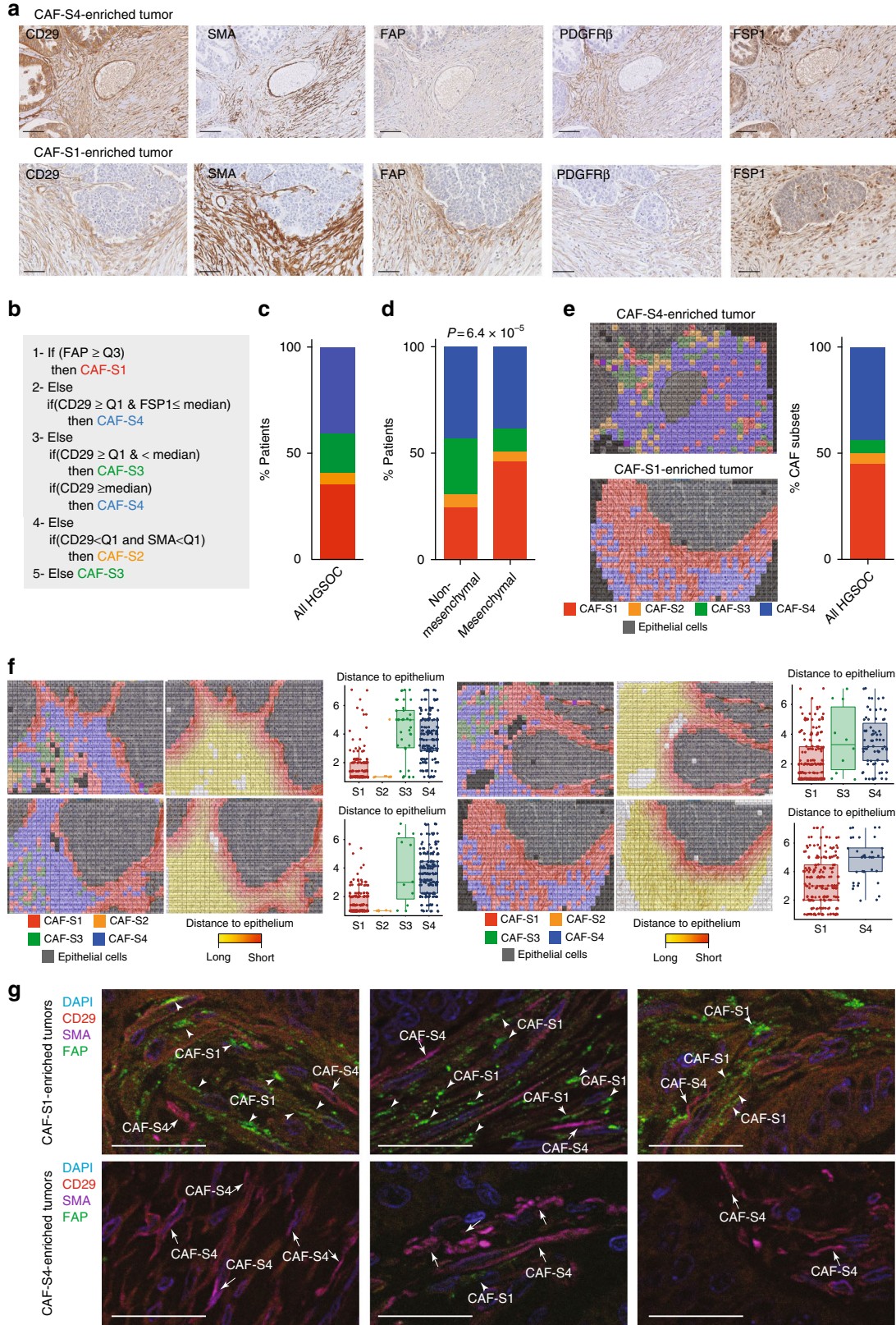

(Fig. 2e and Supplementary Fig. 1b). In addition, we observed on these maps that there was a significant enrichment of CAF-S1 cells at close proximity of cancer cells (Fig. 2f), suggesting a mutual potential benefit between CAF-S1 fibroblasts and cancer cells, as recently shown in pancreatic cancers[26]. Finally, we confirmed the identity of CAF-S1 and CAFS4 cells by using triple immunofluorescence staining of CD29, SMA, and FAP markers on the same HGSOC sections (Fig. 2g). Thus, IHC data (obtained from a retrospective cohort of HGSOC patients) confirm FACS observations described above on fresh HGSOC samples and demonstrate for the first time the existence of four distinct CAF sub-populations in HGSOC, with strong accumulation of the CAF-S1 subset in mesenchymal HGSOC.

**FOXP3$^+$ T lymphocytes accumulate in HGSOC enriched in CAF-S1.** By looking at several HGSOC tumor bed sections, we observed that the stromal compartment accumulated more lymphocytes than the epithelial compartment (see representative hematoxylin−eosin sections, Fig. 3a). This led us to hypothesize that fibroblasts could play a key role in tumor-infiltrating lymphocyte (TIL) recruitment. We thus evaluated T lymphocytes density by performing CD3, CD8, and FOXP3 IHC staining and counted the number of TILs per surface of stromal and epithelial compartments (Fig. 3b–d; Supplementary Fig. 2a for isotype controls). We validated that CD3$^+$, CD8$^+$, and FOXP3$^+$ T lymphocytes were indeed more often detected at the surface of stroma than of epithelium compartment (Fig. 3b–d), thereby highlighting the potential role of CAF on T lymphocytes infiltration within tumor bed. We thus sought to investigate the role of the two most detected fibroblast subsets in HGSOC, CAF-S1, and CAF-S4, on TIL recruitment, by comparing CAF-S1- and CAF-S4-enriched tumors (Fig. 3e–j). While the density of CD31$^+$ blood vessel was similar in CAF-S1- and CAF-S4-enriched HGSOC (Supplementary Fig. 2b, c), the global content in CD3$^+$ lymphocytes was higher in CAF-S1-enriched HGSOC than in CAF-S4-enriched tumors (Fig. 3e, f). This effect was driven by the stroma, as no difference was observed in the epithelium (Fig. 3f, right). In contrast to CD3$^+$ T lymphocytes, no significant difference in the total number of CD8$^+$ cells was observed between CAF-S1- and CAF-S4-enriched tumors (Fig. 3g, h), although they tended to accumulate in CAF-S1-enriched stroma (Fig. 3h, right). Interestingly, the most striking difference between CAF-S1- and CAF-S4-enriched HGSOC was observed with FOXP3$^+$ T lymphocytes (Fig. 3i, j). Indeed, FOXP3$^+$ T cells strongly accumulated in CAF-S1-enriched HGSOC (Fig. 3j), and this enrichment was only seen in the stromal compartment (Fig. 3j). As the total surface of stroma was larger in CAF-S1-enriched HGSOC than in CAF-S4-enriched tumors (as shown in Fig. 1), we normalized the number of CD3$^+$ and FOXP3$^+$ T lymphocytes per unit surface

area of stroma (Fig. 3k, l) and observed that the content in CD3$^+$ and FOXP3$^+$ T cells remained significantly higher in the stroma of CAF-S1-enriched HGSOC, independently of the stromal content (Fig. 3k, l). Finally, we took advantage of the CAF map built on HGSOC (as shown in Fig. 2e) and calculated the distance between CAF subsets and CD3+ T lymphocytes (Fig. 3m). By comparing the number of CD3$^+$ cells at the surface of each CAF subset within HGSOC sections, we confirmed that the proportion of CD3$^+$ T lymphocytes was higher at the surface of CAF-S1-cells compared to the other CAF subsets (Fig. 3m). Altogether, these observations show that HGSOC enriched in CAF-S1 fibroblasts are highly infiltrated in particular by FOXP3$^+$ T lymphocytes.

***CXCL12β* expression discriminates CAF-S1 from CAF-S4.** In order to uncover CAF-S1-mediated functions in T-cell recruitment in HGSOC, we compared CAF-S1 and CAF-S4 transcriptomic profiles by performing RNA sequencing (RNA-Seq) on sorted cells from fresh HGSOC using the gating strategy shown in Fig. 1 (CAF-S1 and CAF-S4 RNAseq data from HGSOC are available using EBI accession number: EGAS00001002184). Unsupervised principal component analysis (PCA) (Fig. 4a) and hierarchical clustering (HC) (Fig. 4b) of the 500 most variant transcripts revealed molecular differences between CAF-S1 and CAF-S4 subsets in HGSOC. DAVID analysis (https://david. ncifcrf.gov), using GO (Gene ontology) and KEGG (Kyoto Encyclopedia of Genes and Genomes) databases, of the differentially expressed genes (CAF-S1-specific gene signature provided in Supplementary Data 1) showed that CAF-S1 cells were enriched in genes involved in biological adhesion, wound healing response, extracellular matrix (ECM) protein remodeling, and skeletal system (Table 2), while CAF-S4 gene signature pinpointed muscle contraction and blood vessel development (Table 3). Interestingly, genes expressed in CAF-S1 cells included many genes enriched in the mesenchymal HGSOC subtype (Fig. 4c), thereby confirming that CAF-S1 fibroblasts could be key components in this HGSOC molecular subtype. The common genes between CAF-S1 and mesenchymal signatures encoded ECM components and proteins involved in immune regulation, such as complement factors (*C1S* and *CFH*), cytokines (*TNFSF4*), and chemokines (*CXCL12β*) that could be involved in T-cell recruitment. We got particularly interested in the detection of the CXCL12 beta isoform (*CXCL12β*), as a transcript significantly upregulated in CAF-S1 subset. Indeed, expression of *CXCL12β* was specific of the CAF-S1 fibroblasts and almost undetected in CAF-S4 fibroblasts. (Fig. 4d). This differential expression was specific of *CXCL12β*. Indeed, *CXCL12α* was expressed at similar levels in both CAF-S1 and CAF-S4 subsets (Fig. 4d, right). In addition, genes that were either positively- or negatively correlated with expression of the *CXCL12β* isoform completely

---

**Fig. 2** Mesenchymal HGSOC accumulate mostly the CAF-S1 subset. **a** Representative views of CD29, SMA, FAP, PDGFRβ, and FSP1 immunostaining of serial sections in CAF-S4- or CAF-S1-enriched HGSOC. Scale bar, 100 μm. **b** Decision tree used to define CAF identity, based on four equal quartiles (Q) and median (Mdn) distribution of each CAF marker intensity. Thresholds (Mdn, Q) and order of decisions were first established from FACS data of a prospective cohort of HGSOC patients ($N = 22$) and next transposed to values of IHC data, using a learning set of tumors containing both non-activated and activated CAF ($N = 60$). **c** Bar plot showing percentage of HGSOC according to the predominant CAF subset detected in each tumor. CAF enrichment per tumor is defined by applying the histological scores of all markers on the decision tree described in (**b**). HGSOC enriched in CAF-S1 (red), CAF-S2 (orange), CAF-S3 (green), or CAF-S4 (blue) are shown as percentage (%). $N = 118$ HGSOC patients. **d** Same as in (**c**) considering mesenchymal ($N = 66$) versus non-mesenchymal ($N = 49$) HGSOC. P values are from Fisher's exact test. **e** Maps of CAF subsets at cellular level, corresponding to the tumor sections shown in (**a**). Each square of 225 μm$^2$ corresponded on average to a single cell. Each CAF subset is represented by a color code and epithelial cells are in grey. The bar plot shows the percentage of HGSOC according to the predominant CAF subset evaluated on CAF maps at cellular level ($N = 9$). **f** Representative views of CAF maps, with their corresponding heatmaps showing the distances (shortest in red, farthest in yellow) between cancer cells and CAF subsets. Scatter plots show the distance to epithelial cells according to CAF subsets (distance calculated in a maximum area of five successive tiles in x and y). Data are shown as mean ± SEM ($n = 425$ cells per image in average). P values are from Student's t-test. **g** Representative images showing triple immunofluorescence co-staining of CD29 (red), FAP (green), and SMA (violet) markers in HGSOC enriched in CAF-S1 (arrowheads) or CAF-S4 (arrows). Scale bar, 50 μm

**Table 1 Comparative description of the clinical parameters of the Institut Curie, AOCS and TCGA cohorts of HGSOC patients**

|  | CURIE | AOCS | TCGA | CURIE |
|---|---|---|---|---|
| Type of analysis | Transcriptomic | Transcriptomic | Transcriptomic | IHC |
| Number of patients | 107 | 285 | 484 | 118 |
| Date of inclusion | 1989−2005 | 1992−2006 |  | 1994−2011 |
| *Age at diagnostic* |  |  |  |  |
| Median age (years) | 58 | 59 | 59 | 60 |
| Range (years) | 31−87 | 22−80 | 30−87 | 35−80 |
| *Histotype* |  |  |  |  |
| Serous | 82 (76.5%) | 264 (92.6%) | 484 (100%) | 115 (97.4%) |
| Endometrioïd | 8 (7.5%) | 20 (7%) |  | 2 (1.7%) |
| Mucinous | 8 (7.5%) |  |  | 1 (0.8%) |
| Clear cell | 6 (5.5%) |  |  |  |
| Carcinosarcoma | 2 (2%) |  |  |  |
| Brenner tumor | 1 (1%) |  |  |  |
| Adenocarcinoma |  | 1 (0.4%) |  |  |
| *Figo substage* |  |  |  |  |
| I | 21 (19.6%) | 24 (8.4%) |  | 7 (5.9%) |
| II | 10 (9.35%) | 18 (6.3%) | 24 (5%) | 9 (7.6%) |
| III | 59 (55.14%) | 217 (76.1%) | 377 (77.9%) | 82 (69.5%) |
| IV | 17 (15.9%) | 22 (7.7%) | 78 (16.1%) | 13 (11.01%) |
| NA |  | 4 (1.4%) | 5 (1%) | 7 (5.9%) |
| *Grade* |  |  |  |  |
| 1 | 7 (6.5%) | 19 (6.7%) |  |  |
| 2 | 34 (31.5%) | 97 (34%) | 57 (11.8%) | 31 (26.3%) |
| 3 | 66 (62%) | 164 (57.5%) | 415 (85.7%) | 87 (73.7%) |
| NA |  | 5 (1.8%) | 12 (2.5%) |  |
| *Surgery* |  |  |  |  |
| Full | 38 (36%) | 84 (29.5%) | 88 (18.2%) | 32 (27.1%) |
| Partial | 69 (64%) | 164 (57.5%) | 339 (70%) | 82 (69.5%) |
| NA |  | 37 (13%) | 57 (11.8%) | 4 (3.4%) |
| *Clinical response* |  |  |  |  |
| RC—Complete response | 51 (47.7%) |  | 273 (56.4%) | 47 (39.8%) |
| RP—Partial response | 22 (20.6%) |  | 57 (11.8%) | 31 (26.3%) |
| S—Stability | 7 (6.5%) |  | 25 (5.2%) | 7 (5.9%) |
| P—Progression | 11 (10.3%) |  | 36 (7.4%) | 3 (2.6%) |
| NA | 16 (15%) |  | 93 (19.2%) | 30 (25.4%) |

The first three columns recapitulate clinical parameters from cohorts used for transcriptomic data analyses. TCGA, AOCS and Curie cohorts have previously been described[6–8]. The last column concerns samples used for immunohistochemistry analyses. Tumor samples were obtained from a cohort of consecutive ovarian carcinoma patients, treated at the Institut Curie between 1989 and 2012. All analyzed samples have been collected prior to any chemotherapeutic treatment. Indeed, for each patient, a surgical specimen was taken, before chemotherapy, for pathological analysis and tumor tissue cryopreservation. The median patient's age was 60 years (with a range of 35–80 years). Ovarian carcinomas were classified according to the World Health Organization histological classification of gynecological tumors. Pathological analysis identified 115 high-grade serous tumors (97.4%), 2 high-grade endometrioïd tumors (1.7%) and 1 high-grade mucinous tumor (0.8%). Sixteen subjects (13.5%) were considered as early stage (International Federation of Gynecology and Obstetrics (FIGO) I−II) and 95 subjects (80.5%) were considered as advanced stage (III and IV) of disease. Patients were treated with a combination of surgery and chemotherapy, the latter including alkylating or alkylating-like agents ± taxane as a first-line treatment in most cases. All the subjects underwent surgery, 82 of them have a partial debulking and 32 subjects have a full debulking

recapitulated CAF-S1 and CAF-S4 genetic signatures, respectively (Fig. 4e). In contrast, this was not the case for *CXCL12α* (Fig. 4f), thus confirming that only the *CXCL12β* isoform is discriminant between CAF-S1 and CAF-S4 cells. As CXCL12 prognostic value was highly controversial in HGSOC, with variable impacts on patient survival[22–24], we hypothesized that the isoform-specific regulation of CXCL12 in CAF subsets could be of particular interest. We thus analyzed micro-arrays data from three independent HGSOC cohorts (Curie, AOCS, and TCGA) (Table 1) and showed that high *CXCL12β* mRNA level was invariably associated with poor patient survival (Fig. 4g, for Curie Cohort; Supplementary Fig. 3a, for AOCS and TCGA cohorts, and Supplementary Fig. 3c for analyses by iteration). In contrast, *CXCL12α* was not a reliable prognostic factor in HGSOC (Supplementary Fig. 3b). Importantly, when we considered the expression of the two detected isoforms together (referred to as total CXCL12 expression), we observed that the prognostic value of total CXCL12 followed the one of *CXCL12β* (Supplementary Fig. 3e), arguing for the important role of the *CXCL12β* isoform in HGSOC. Consistent with the poor prognosis associated with high *CXCL12β* mRNA levels, we observed that *CXCL12β* expression was higher in mesenchymal HGSOC compared to

non-mesenchymal HGSOC (Fig. 4h). This was further validated using contingency tables based on data from the three cohorts of patients that showed a strong enrichment of HGSOC with high *CXCL12β* mRNA levels among the mesenchymal HGSOC (Supplementary Fig. 3f). *CXCL12α* expression was also detected in mesenchymal HGSOC (Supplementary Fig. 3g), but the balance between *CXCL12β* and *CXCL12α* expression—assessed by the ratio of *CXCL12β* to *CXCL12α* mRNA levels—was significantly in favor of the *CXCL12β* isoform in mesenchymal HGSOC (Fig. 4i). Finally, we studied CXCL12/CXCR4 protein expression patterns in HGSOC by IHC (Fig. 4j–l). As there is no available antibody recognizing specifically the *CXCL12β* isoform, we had to use an antibody recognizing both CXCL12 isoforms[24,27], for which we validated the specificity (Supplementary Fig. 4a−c). As expected, we confirmed that CXCL12 proteins were mainly detected in the stroma (Fig. 4j), and its receptor CXCR4 at the surface of epithelial cells (Fig. 4k, arrows). CXCR4 was also detected in endothelial and immune cells (Fig. 4k, arrowheads), underlying the role of the CXCL12/CXCR4 axis in HGSOC microenvironment. We observed a huge diversity in CXCL12 histological scoring in HGSOC (Fig. 4l). Still, CXCL12 protein significantly accumulated in mesenchymal HGSOC (Fig. 4l), thereby

confirming transcriptomic data. In contrast, no difference was observed for CXCR4 protein levels between mesenchymal and non-mesenchymal HGSOC (Fig. 4l, right). Finally, by performing in situ hybridization using the RNAscope® technology, we observed that *CXCL12* mRNA was mainly expressed by stromal cells (Fig. 4m), observation that was confirmed by using publicly

available data set from primary cell lines (GSE49910 from Gene Expression Omnibus resources) (Supplementary Fig. 5). Taken as a whole, these data indicate that the *CXCL12β* isoform accumulates more than *CXCL12α* in mesenchymal HGSOC, consistent with the accumulation of CAF-S1 fibroblasts in these tumors. All these features, i.e. mesenchymal molecular subtype, CAF-S1

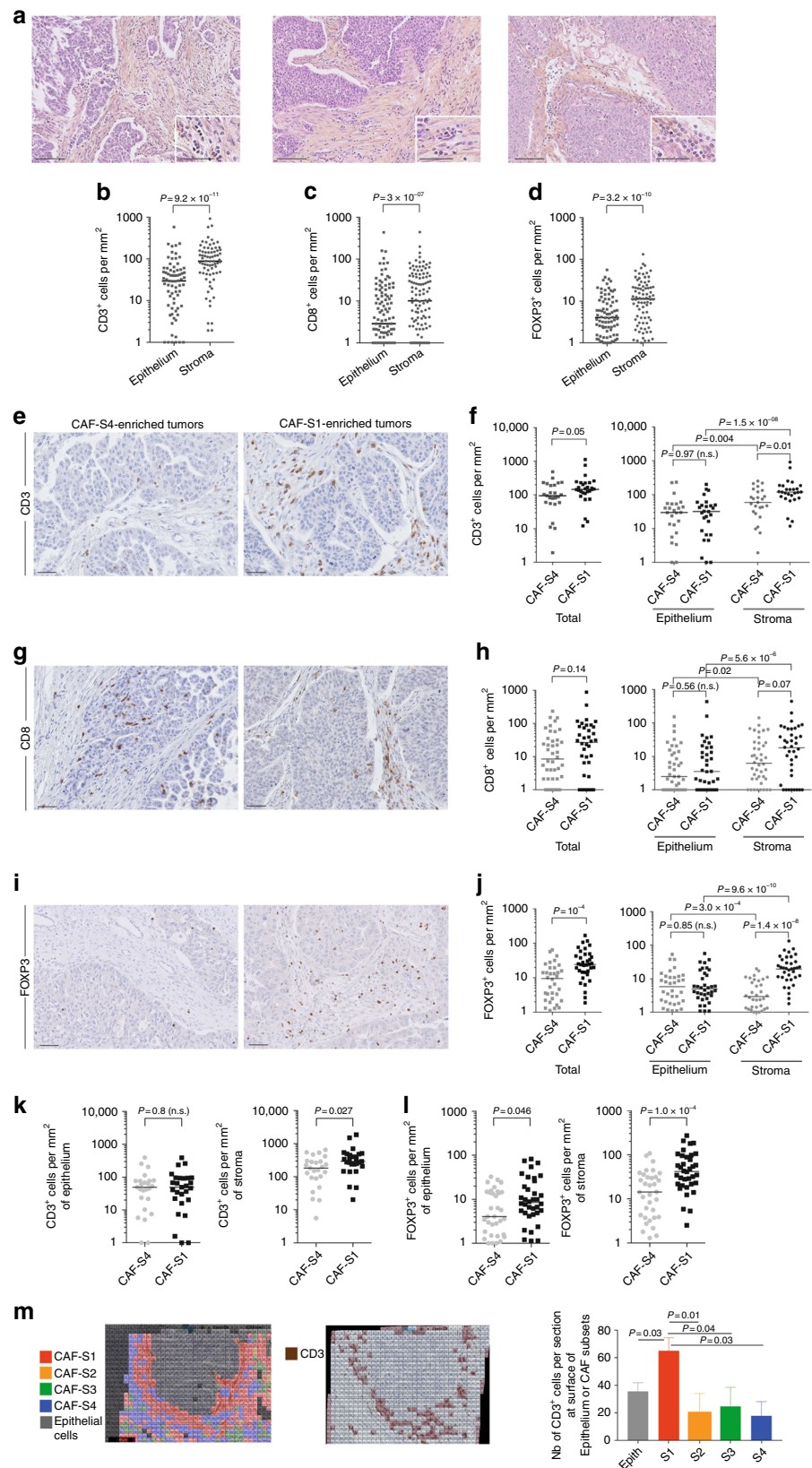

accumulation and *CXCL12β* expression, are hallmarks of dismal prognosis for HGSOC patients.

**CAF-S1 enhance regulatory T-cell activity at tumor site**. As regulatory T lymphocytes preferentially infiltrate CAF-S1-enriched stroma, we wondered whether CAF-S1 subset expressing high levels of *CXCL12β* could favor FOXP3+ T lymphocytes accumulation by increasing their attraction, enhancing their survival and/or promoting their activation state, all being non-exclusive hypotheses. We investigated the impact of CAF-S1 fibroblasts on T-cell attraction by performing in vitro transwell assays. To do so, we isolated CAF-S1 and CAF-S4 fibroblasts from ovarian tumors. While CAF-S1 fibroblasts proliferated well and kept their properties in vitro (Supplementary Fig. 6a–d), CAF-S4 fibroblasts quickly died and could not be maintained in culture, thereby precluding any comparison between CAF-S1 and CAF-S4 fibroblasts in functional assays. Still, we succeeded in isolating CAF-S1 primary cells that enabled us to analyze in vitro the impact of primary CAF-S1 fibroblasts on CD4+CD25− and CD4+CD25+ T cells isolated from peripheral blood mononuclear cells (PBMC) of healthy donors (Supplementary Fig. 6e). We first observed that migration of CD4+CD25+ but not CD4+CD25− cells was increased in presence of primary CAF-S1 fibroblasts (Fig. 5a), while the survival of the two types of T lymphocytes was not affected by distant CAF-S1 fibroblasts (Fig. 5b). Consistent with CXCL12 function in this process, CXCR4 receptor was detected at higher levels at the surface of CD4+CD25+ T cells than CD4+CD25− T lymphocytes (Fig. 5c, d). Moreover, CXCR4 level at the surface of CD4+CD25+ T lymphocytes was increased upon co-culture with CAF-S1 fibroblasts (Fig. 5c, d, right), suggesting that the CXCL12/CXCR4 axis could be activated in T-lymphocytes in presence of CAF-S1 fibroblasts and required for CAF-S1-induced CD4+CD25+ T-lymphocyte attraction. As CAF-S1 cells specifically accumulate the *CXCL12β* isoform, but also express *CXCL12α*, we next silenced each of these isoforms in CAF-S1 fibroblasts (Fig. 5e) to address their respective function in CD4+CD25+ cell attraction. The silencing of *CXCL12α* decreased T-cell attraction (Fig. 5f), confirming previous reports[28,29]. Interestingly, *CXCL12β* knockdown in CAF-S1 cells, which mimicked CAF-S4 cells as they express *CXCL12α* but not *CXCL12β*, significantly reduced CD4+CD25+ T-lymphocyte attraction (Fig. 5f). Importantly, silencing both isoforms showed an additive effect and reduced T-cell attraction that returned to basal level (i.e. without CAF-S1 (Fig. 5f)). Thus, co-silencing of the two isoforms was both necessary and sufficient to completely abrogate T-lymphocyte attraction by CAF-S1 cells (Fig. 5f), indicating that in addition to *CXCL12α*, the specific expression of *CXCL12β* in CAF-S1 is absolutely essential for efficient T-lymphocyte attraction. While expression of chemokines, such as CCL17, CCL22, CCL5, CCL28, and CXCL9, well known to be involved in recruitment of regulatory T lymphocytes[30] was not detected in CAF-S1, CCL2 expression was highly expressed by

CAF-S1 cells (RNAseq data available EGAS00001002184, Supplementary Data 1 for CAF-S1 signature). We thus tested whether this other chemokine could be involved in T-cell attraction by CAF-S1, besides CXCL12, and found that, in contrast to CXCL12, CCL2 inhibition in CAF-S1 fibroblasts (Supplementary Fig. 6f for silencing efficiency) had no impact on CD4+CD25+ T-lymphocyte attraction (Fig. 5g).

To get further insights on the impact of CAF-S1 cells on CD4+CD25+ T lymphocytes, we next performed co-culture experiments (Fig. 5h–m). We observed that the direct contact of CAF-S1 fibroblasts with CD4+CD25+ T cells significantly increased the proportion of CD25+FOXP3+ T cells among CD4+ T lymphocytes (Fig. 5h, i). Moreover, CAF-S1 cells enhanced the survival of CD25+FOXP3+ T lymphocytes (Fig. 5j). Direct contact of CD25+FOXP3+ T lymphocytes with CAF-S1 was required for this effect on T-cell survival, as no impact was observed in Transwell assays (as shown above Fig. 5a). Importantly, the increase in the number of CD25+FOXP3+ T cells by CAF-S1 cells was conserved when it was reported to the survival (Fig. 5k), suggesting that CAF-S1 fibroblasts increase the global content of CD25+FOXP3+ T cells by enhancing their survival and differentiation. These effects were not affected by the combined silencing of the two CXCL12 isoforms in CAF-S1 (Fig. 5h–k), indicating that the two CXCL12 isoforms are required for attracting CD4+CD25+ T lymphocytes, but not for enhancing survival or differentiation into CD25+FOXP3+ T cells. Considering the increase of CD25+FOXP3+ T cells upon co-culture with CAF-S1 fibroblasts, we next sought to verify if this effect was associated with an increased T-cell suppressive activity. To do so, we isolated CD4+CD25^HighCD127^lowCD45RA^low T lymphocytes, strongly enriched in regulatory T cells, and evaluated their impact on CD4+ effector T cells following co-culture with CAF-S1 fibroblasts (Fig. 5l). Interestingly, we found that the pre-culture of CD25^HighCD127^lowCD45RA^low T lymphocytes with CAF-S1 fibroblasts significantly enhanced their capacity to inhibit effector T-cell (CD4+CD25−) proliferation rate (Fig. 5l). Consistent with the increase in CD25+FOXP3+ T cells, these data suggest that CAF-S1 cells could increase immunosuppressive activity of regulatory T lymphocytes. Finally, to get better insights in the mechanisms mediated by CAF-S1 cells on the differentiation of CD25^HighFOXP3^High T cells, we took advantage of the CAF-S1 RNAseq data (EGAS00001002184) and identified different molecules highly expressed by CAF-S1 cells that could be involved in this effect. Among them, we observed that the silencing of CD73/NT5E, B7H3/CD276 and IL6 in CAF-S1 fibroblasts (Supplementary Fig. 6f for silencing efficiency) significantly reduced the proportion of CD25^HighFOXP3^High T-lymphocyte population (Fig. 5m, n). Taken as a whole, these data indicate that, in addition to *CXCL12α*, the expression of *CXCL12β* by CAF-S1 fibroblasts is essential for T-cell attraction towards CAF-S1-enriched HGSOC. Once in contact, CAF-S1

**Fig. 3** CAF-S1-enriched HGSOC accumulate FOXP3+ T lymphocytes. **a** Representative views of HES staining of HGSOC tumor bed sections (Institut Curie cohort) showing lymphocytes accumulation at the surface of the stroma. Scale bar, 100 μm (low magnification) and 50 μm (inset). **b−d** Scatter plots showing the number of CD3+ (**b**), CD8+ (**c**), and FOXP3+ (**d**) lymphocytes per mm² in epithelial and stromal compartments in HGSOC. $N = 80$ HGSOC (Institut Curie). $P$ values are from Wilcoxon signed-rank test. **e, g, i** Representative views of CD3+ (**e**), CD8+ (**g**), and FOXP3+ (**i**) immunostaining in HGSOC enriched in CAF-S4 or CAF-S1. Scale bar, 50 μm. **f, h, i** Number of immune cells per mm² in CAF-S1- and CAF-S4-enriched HGSOC, considering either the whole sections referred to as Total or the epithelial and stromal compartments. Positive cells for each staining were counted manually in at least 5−10 fields per tumor at ×20 magnification. The median is indicated. Mann-Whitney statistical test was performed to compare CAF-S1- versus CAF-S4-enriched tumors and Wilcoxon paired test was used to compare epithelial and stromal compartments within tumors. $N = 80$ HGSOC (Institut Curie). **k, l** Scatter plots showing the number of CD3+ (**k**) and FOXP3+ (**j**) T lymphocytes per mm² relative to the stromal or epithelial content per tumor, in CAF-S1- or CAF-S4 enriched HGSOC. $P$ values are from Mann-Whitney test. **m** Representative views of CAF maps, with the corresponding heatmap showing the localization of CD3+ T lymphocytes (0.25 mm²). The bar plot shows the number of CD3+ T lymphocytes detected at the surface of epithelial cells (Epith) or of each CAF subset cell ($n = 936$ total cells per image, five images from different HGSOC were analyzed). $P$ values are from Student's $t$-test

fibroblasts enhance the survival and the proportion of CD25⁺FOXP3⁺ T lymphocytes, through at least CD73, B7H3 and IL-6.

**CXCL12β RNA is targeted by miR-141/200a in CAF-S4 cells.** As *CXCL12β* expression in CAF-S1 is essential for T-cell attraction, we next wondered the mechanism driving its specific

expression in CAF-S1 cells, compared to CAF-S4 cells. We considered the possibility that *CXCL12β* could be downregulated in CAF-S4 cells by miRNA. Indeed, gene signatures differentiating mesenchymal (high CAF-S1 content) to non-mesenchymal HGSOC (high CAF-S4 content) were based on miR-141 and 200a family members[7,13]. We thus hypothesized that *CXCL12β*

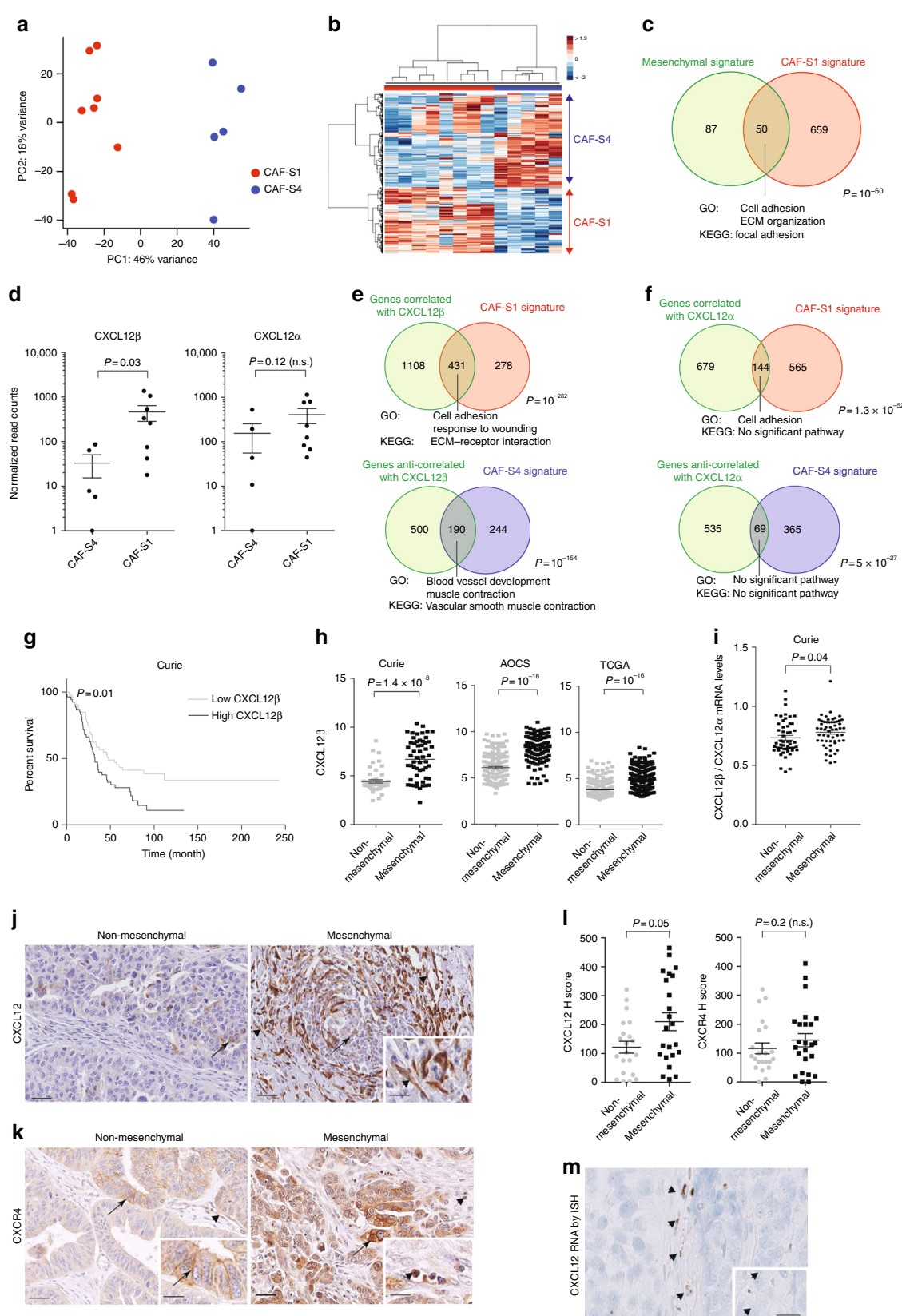

mRNA could be targeted by miR-141/200a in CAF-S4 cells, leading to its specific accumulation in CAF-S1 fibroblasts. We identified two predicted miR-141/200a binding sites in the 3′-UTR of *CXCL12β* mRNA, which were absent in *CXCL12α* mRNA (Fig. 6a). We first observed that the *CXCL12β* 3′-UTR contained genuine miR-141/200a binding sites (Fig. 6b). Moreover, over-expression of miR-141 or miR-200a in CAF-S1 fibroblasts reduced the total level of endogenous *CXCL12β* mRNA, but had no impact on *CXCL12α* (Fig. 6c). Furthermore, the *CXCL12β* mRNA level was inversely correlated with the miR-141 and miR-200a in HGSOC (Fig. 6d). Consistently, miR-141 and miR-200a significantly accumulated in non-mesenchymal HGSOC (Fig. 6e), which exhibited low levels of *CXCL12β* mRNA. We and others have demonstrated that the miR-200 family members are strongly upregulated by oxidative stress in various cell types, including fibroblasts[7,13,31–34]. We thus wondered whether non-mesenchymal HGSOC, accumulating CAF-S4 fibroblasts, could suffer from oxidative stress. We compared the amount of reduced (GSH) and oxidized (GSSG) glutathione in non-mesenchymal and mesenchymal HGSOC (Fig. 6f). We observed that non-mesenchymal HGSOC significantly accumulated more GSSG compared to the mesenchymal ones (Fig. 6f). Accordingly, while GSSG and GSH showed similar proportion in mesenchymal HGSOC, the proportion of GSSG tended to be significantly higher than GSH in non-mesenchymal HGSOC (Fig. 6f, right). Using Gene Set Enrichment Analysis, we detected a strong enrichment in genes encoding electron transport chain proteins in CAF-S4 RNAseq data compared to CAF-S1, confirming an oxidative metabolism in CAF-S4 cells (Fig. 6g). The small amount of material obtained from CAF-S1 and CAF-S4 cells sorted by FACS from HGSOC precluded direct measurement of miR-200 levels in these cells. We thus took advantage that expression of the miR-141 is strikingly correlated with the transcription of its immediate upstream gene *PTPN6*[13], and used *PTPN6* as a read-out of miR-141 expression level in CAF-S1 and CAF-S4. *PTPN6* expression was significantly higher in CAF-S4 compared to CAF-S1 fibroblasts (Fig. 6h), suggesting that miR-141/200c expression was also upregulated in CAF-S4. Accordingly, three previously identified targets of miR-141/200a[7,35,36] showed a significant downregulation in CAF-S4 fibroblasts compared to CAF-S1 (Fig. 6i), similarly to *CXCL12β*. Taken as a whole, these data established that the *CXCL12β* isoform is specifically targeted by the miR-200 family members, while *CXCL12α* is not. The miR-200 are upregulated in non-mesenchymal HGSOC enriched in CAF-S4 fibroblasts. Subsequently, the *CXCL12β* isoform is specifically downregulated in CAF-S4 fibroblasts, while it accumulates in CAF-S1 cells, further enhancing attraction of CD4$^+$CD25$^+$ T lymphocytes (see model Fig. 6j).

## Discussion

Here, we uncover CAF heterogeneity in HGSOC by identifying four different CAF subpopulations, including two myofibroblast subsets, referred to as CAF-S1 (CD29$^{Med-Hi}$ FAP$^{Hi}$ SMA$^{Med-Hi}$ FSP1$^{Med-Hi}$ PDGFRβ$^{Med-Hi}$ CAV1$^{Low}$) and CAF-S4 (CD29$^{Hi}$ FAP$^{Low}$ SMA$^{Hi}$ FSP1$^{Hi}$ PDGFRβ$^{Med-Hi}$ CAV1$^{Neg-Low}$). Mesenchymal HGSOC, the molecular subgroup of ovarian cancers with poor patient prognosis, exhibit high content of CAF-S1 fibroblasts. CXCL12 is required for CAF-S1-mediated T-lymphocyte attraction in HGSOC, consistent with previous observations in pancreatic cancers[37,38]. In addition, we highlight that the *CXCL12β* isoform, but not *CXCL12α*, is a key component for differentiating the CAF-S1 from the other activated CAF-S4 fibroblasts in HGSOC. This specific expression of *CXCL12β* in CAF-S1 is essential for T-cell attraction towards CAF-S1-enriched HGSOC, and thus could be required for CAF-S1 immunosuppressive function. Once in contact, CAF-S1 fibroblasts enhance the survival, as well as the activation of regulatory T lymphocytes, independently of CXCL12. Finally, we uncover the mechanism driving the specific expression of CXCL12β in CAF-S1. The CXCL12β isoform is regulated by an miR-141/200a-dependent mechanism that downregulates *CXCL12β* expression in CAF-S4 fibroblasts and promotes its specific accumulation in the CAF-S1 subpopulation (see model, Fig. 6j).

Based on genetic and transcriptomic analyses, it is now well established that HGSOC are composed of heterogeneous molecular entities. All studies based on ovarian cancer transcriptomic profiles[6–13] systematically identified a group of HGSOC defined by a mesenchymal signature that is invariably associated with poor patient prognosis. This signature contains many stromal-related genes[6–13], suggesting that the stroma could be an important feature for the aggressiveness of these tumors. Interestingly, both SMA and PDGFRβ markers, used in our study, belong to several mesenchymal signatures[6,7]. While high proportion of CAF is associated with poor prognosis in HGSOC patients[17,18,20,39–41], little is known about their identity. Some studies have analyzed markers, such as SMA, FAP or PDGFRβ, individually and show a certain degree of heterogeneity in ovarian cancers[15,42–44]. Our current study confirms these observations but goes beyond by identifying heterogeneous CAF subsets and characterizing them at histological and molecular levels. To do so, we combined the use of six different markers, previously tested individually but never studied concomitantly[45–48]. To our knowledge, the present study is the first that identifies four different CAF subpopulations in HGSOC, including CAF-S1 and CAF-S4, which express high levels of polymerized SMA and can be defined as myofibroblasts. Still, they exhibit different transcriptomic profiles, arguing for distinct functions.

**Fig. 4** *CXCL12β* discriminates CAF-S1 and CAF-S4 cells. **a** PCA based on the 500 most variant transcripts differentiating CAF-S1 (red) and CAF-S4 (blue). **b** HC (500 most variant transcripts) using Ward's method with Euclidean distances. Each column represents a CAF subset and each row a gene. Color saturation shows gene expression deviation from the mean (above in red, below in blue). **c** Venn diagram showing overlap between mesenchymal signature (defined in ref. [7]) and CAF-S1 signature (Supplementary Data 1). *P* value is from hypergeometric test. **d** Scatter plots of *CXCL12α* (NM_000609) or *CXCL12β* (NM_199168) mRNA levels in CAF-S1 and CAF-S4 subsets. **e, f** Venn diagrams showing overlap between genes correlated or anti-correlated with *CXCL12β* (**e**) or *CXCL12α* (**f**) and CAF-S1 or CAF-S4 signatures (Supplementary Data 1 and 2). *P* values are from hypergeometric test. **g** Kaplan−Meier curves of overall survival according to low- and high-*CXCL12β* mRNA levels ($N = 53$ in low-*CXCL12β* subgroup and $N = 54$ patients in high-*CXCL12β* subgroup, Institut Curie). *P* value is based on log-rank test. **h** Scatter plots showing *CXCL12β* mRNA levels in mesenchymal and non-mesenchymal HGSOC from the Institut Curie, AOCS, and TCGA cohorts. Data (log2 of probeset (203666_at) intensity) are shown as mean ± SEM. *P* values are from Mann-Whitney test. **i** Scatter plot showing ratio of *CXCL12α* and *CXCL12β* expression levels in mesenchymal and non-mesenchymal HGSOC of the Institut Curie cohort. Data are shown as mean ± SEM. *P* values are from Mann-Whitney test. **j, k** Representative views of CXCL12 (**j**) and CXCR4 (**k**) immunostaining in HGSOC. Scale bar, 50 μm (low magnification) and 20 μm (inset). **l** Scatter plots showing histological scores (H score) of CXCL12 and CXCR4 proteins. H score corresponds to the percentage of positive cells (in CAF and at epithelial cell surface, arrows in **j, k**) multiplied by the staining intensity. Data are shown as mean ± SEM. *P* values are from Mann-Whitney test. **m** Representative view of *CXCL12* mRNA detected in fibroblasts by in situ hybridization, using RNAscope® Technology on HGSOC tissue section. Scale bar, 20 μm (low magnification) and 6 μm (inset)

**Table 2 Significant enriched pathways for genes upregulated in CAF-S1 versus CAF-S4**

| GO_Biological Process-Term | Count | % | Genes | *P* value | FDR |
|---|---|---|---|---|---|
| GO:0007155~cell adhesion | 79 | 11.67 | NRP2, AEBP1, TLN2, CXCL12, CDSN, SDC3, CHAD, WISP1, CTGF, COL12A1, ROBO2, COL11A1, BOC, NEGR1, CDH23, SPON1, CYR61, PDPN, CDHR1, SIGLEC11, PCDH7, CERCAM, JUP, CD34, LSAMP, CPXM1, CLDN1, CNTN1, ROR2, VCAN, JAM2, CHL1, CLDN18, PLXNC1, COL3A1, PTK7, NINJ2, ITGB5, SPOCK1, CDH5, DCHS1, ITGBL1, ISLR, IGSF11, ANXA9, FAT4, ITGB8, COMP, COL6A3, SCARB1, COL8A1, COL8A2, THBS2, MLLT4, FLRT2, SVEP1, LRRN2, PPFIBP1, NLGN1, HSPG2, COL16A1, TPBG, COL5A1, COL4A6, EMILIN1, CCL11, LAMA1, OMD, DSG2, PKP2, CDON, FBLN5, DSC3, DSC2, ADAM22, ANTXR1, BMPR1B, NTM, CDH11 | 1.39E-20 | 2.44E-17 |
| GO:0001501~skeletal system development | 36 | 5.32 | RBP4, AEBP1, PTGS2, FGF9, COL3A1, GLI2, MMP2, GLI1, CHAD, VDR, CTGF, COMP, COL12A1, COL11A1, PAPSS2, RUNX2, COL10A1, EVC, FBN1, HSPG2, IGF1, ANKH, HOXC10, PTHLH, SMO, CTSK, COL1A2, PDGFRA, GDF10, ROR2, KIAA1217, COL1A1, BMPR1B, BMP6, CDH11 | 1.72E-9 | 3.01E-6 |
| GO:0030198~extracellular matrix organization | 19 | 2.80 | LUM, COL3A1, HSPG2, CCDC80, DCN, SPINK5, COL5A1, COL4A6, EMILIN1, P4HA1, FBLN5, COL1A2, PDGFRA, COL12A1, LOX, COL1A1, COL11A1, COL8A2, CYR61 | 1.71E-8 | 2.99E-5 |
| GO:0016337~cell-cell adhesion | 29 | 4.28 | CLDN18, NINJ2, CDSN, DCHS1, CDH5, CHAD, ANXA9, FAT4, CTGF, ROBO2, COL11A1, COL8A2, CDH23, PDPN, CDHR1, NLGN1, PCDH7, CERCAM, JUP, DSG2, CD34, PKP2, CLDN1, DSC3, DSC2, ROR2, BMPR1B, JAM2, CDH11 | 3.93E-7 | 6.87E-4 |
| GO:0009611~response to wounding | 41 | 6.05 | C7, TLR1, COL3A1, F2RL1, NINJ2, TLR4, C1S, BDKRB2, GPR68, LMAN1, MDK, CFHR1, HMCN1, SLC1A3, NOD1, CTGF, HMOX1, SERPINE1, CFH, SCARB1, LOX, CFI, PAPSS2, SCG2, NOX4, TNFSF4, PDPN, IGF1, COL5A1, CCL11, SMO, PRKCQ, CD55, SDC1, FBLN5, PDGFRA, VCAN, BMPR1B, GAP43, BMP6, MYH10 | 3.43E-6 | 0.006 |
| GO:0030199~collagen fibril organization | 9 | 1.33 | P4HA1, LUM, COL3A1, COL1A2, COL12A1, COL1A1, LOX, COL11A1, COL5A1 | 4.58E-6 | 0.008 |
| GO:0001944~vasculature development | 25 | 3.69 | NRP2, FGF9, LEPR, COL3A1, FOXO1, CXCL12, MMP2, CDH5, SHB, ANG, CTGF, HMOX1, PLCD3, SEMA3C, HS6ST1, LOX, SCG2, CYR61, PDPN, MMP19, COL5A1, VEGFC, SMO, COL1A2, COL1A1 | 7.47E-6 | 0.01 |
| GO:0001649~osteoblast differentiation | 10 | 1.48 | PTHLH, SMO, FGF9, IGF1, COL1A1, GLI2, IGFBP3, RUNX2, GLI1, BMP6 | 1.11E-5 | 0.02 |
| GO:0000902~cell morphogenesis | 30 | 4.43 | NRP2, SHROOM3, UCHL1, PTK7, GLI2, EPHB3, CXCL12, EPHB2, DAB2, SLC1A3, UNC5B, ROBO2, ROBO3, CDH23, NOX4, EGR2, PDPN, KIF5C, PRKCI, HGF, NTN1, GAS7, SMO, LAMA1, SEMA6A, VCAN, ANTXR1, BMPR1B, GAP43, MYH10 | 1.96E-5 | 0.03 |
| GO:0007411~axon guidance | 15 | 2.21 | NRP2, EGR2, KIF5C, EPHB3, GLI2, CXCL12, NTN1, EPHB2, SEMA6A, UNC5B, ROBO2, ROBO3, BMPR1B, GAP43, MYH10 | 1.99E-5 | 0.03 |
| GO:0035295~tube development | 22 | 3.25 | RBP4, SHROOM3, PDPN, FGF9, PTK7, IGF1, GLI2, CXCL12, GLI1, FOXP2, MYCN, WNT2, PTHLH, WNT4, GPC3, CTGF, PDGFRA, TGIF1, ROBO2, HS6ST1, LOX, CYR61 | 2.80E-5 | 0.04 |
| GO:0006928~cell motion | 35 | 5.17 | NRP2, CTHRC1, IL16, PTGS2, SPOCK1, GLI2, EPHB3, CXCL12, EPHB2, WNT2, UNC5B, CTGF, ANG, SEMA3C, ROBO2, SCARB1, ROBO3, SCG2, EGR2, KIF5C, IGF1, CERCAM, NTN1, SLC9A10, COL5A1, ELMO1, SMO, SEMA6A, LAMA1, VEGFC, CD34, VCAN, BMPR1B, GAP43, MYH10 | 5.61E-5 | 0.09 |

| KEGG_Pathway-Term | Count | % | Genes | *P* value | FDR |
|---|---|---|---|---|---|
| hsa04512:ECM-receptor interaction | 16 | 0.26 | COL3A1, HSPG2, ITGB5, COL5A1, COL4A6, CHAD, SDC3, LAMA1, SDC1, ITGB8, COMP, COL6A3, COL1A2, COL1A1, COL11A1, THBS2 | 4.18E-7 | 4.89E-4 |
| hsa04360:Axon guidance | 18 | 0.30 | ABLIM1, PLXNC1, LIMK2, LIMK1, ABLIM3, LRRC4C, EPHB3, NTN1, CXCL12, EPHB2, SEMA6A, UNC5B, SEMA7A, SRGAP3, SEMA3C, ROBO2, EFNA4, ROBO3 | 5.84E-6 | 6.82E-3 |

Enrichment was performed using DAVID web software based on Gene Ontology and KEGG databases. *P* values are indicated without or after FDR correction

**Table 3 Significant enriched pathways for genes upregulated in CAF-S4 versus CAF-S1**

| GO_Biological Process-Term | Count | % | Genes | P value | FDR |
|---|---|---|---|---|---|
| GO:0001568~blood vessel development | 29 | 7.02 | CAV1, PDGFA, PGF, CSPG4, ENPEP, JAG1, GJA4, PTEN, GJC1, SEMA5A, PTK2, AGT, ANGPT1, ADRA2B, MKL2, FGF1, PLXND1, ANGPT2, COL18A1, FLT1, EPAS1, APOLD1, ITGA4, ARHGAP24, CDH13, LAMA5, PLXDC1, NTRK2, ITGA7 | 5.45E-12 | 9.40E-9 |
| GO:0001525~angiogenesis | 21 | 5.08 | COL18A1, FLT1, EPAS1, PDGFA, PGF, CSPG4, APOLD1, JAG1, ENPEP, ARHGAP24, PTEN, SEMA5A, CDH13, PTK2, LAMA5, PLXDC1, ANGPT1, ADRA2B, PLXND1, FGF1, ANGPT2 | 3.27E-10 | 5.64E-7 |
| GO:0007010~cytoskeleton organization | 35 | 8.47 | DLC1, CAV2, TPPP3, CAV1, PDGFB, CALD1, ARPC5, PRKG1, DAAM2, DSTN, PTK2, EZR, PACSIN2, MICAL1, CAP1, EHD2, FGD4, ARHGDIB, ACTC1, CAP2, ROCK1, MAP1B, ARHGEF17, MYOZ1, VASP, ARHGAP26, MARK1, ARPC1A, PLCE1, EPB41L1, EPS8, LAMA5, MAP2, MYH11, SYNM | 1.10E-9 | 1.90E-6 |
| GO:0007517~muscle organ development | 22 | 5.32 | MEF2C, CAV2, CAV1, MYL4, ACTC1, TBX2, UTRN, RXRG, CACNB2, CSRP2, PTEN, GJC1, FAM65B, MEF2D, LAMA5, PLN, ITGA7, MYH11, RARB, AGRN, MKL2, SGCA | 3.19E-8 | 5.51E-5 |
| GO:0044057~regulation of system process | 25 | 6.05 | CAV1, MYL4, LZTS1, SLC6A1, PPP1R12B, ADA, SYP, EDNRB, KCNQ4, AGT, GUCY1A3, HRC, FLT1, NTF3, EPAS1, MAP1B, ATP1A2, SSTR2, PLCE1, P2RX1, PLN, NTRK2, HSPB7, NPTN, CACNA1C | 3.75E-7 | 6.47E-4 |
| GO:0042692~muscle cell differentiation | 15 | 3.63 | CAV2, ACTC1, NTF3, TBX2, UTRN, CACNB2, MYOZ1, JAG1, CSRP2, AGT, MYH11, SORT1, RARB, AGRN, MKL2 | 1.08E-6 | 0.002 |
| GO:0009190~cyclic nucleotide biosynthetic process | 8 | 1.93 | ADCY3, ADCY4, ADCY1, ADCY5, ADCY6, GUCY1A2, GUCY1A3, GUCY1B3 | 1.35E-6 | 0.002 |
| GO:0042310~vasoconstriction | 7 | 1.69 | ACTG2, EDNRB, CAV1, ACTC1, P2RX1, ACTA2, AGT | 1.72E-6 | 0.003 |
| GO:0009187~cyclic nucleotide metabolic process | 9 | 2.17 | ADCY3, ADCY4, ADCY1, NUDT4, ADCY5, NUDT4P1, ADCY6, GUCY1A2, GUCY1A3, GUCY1B3 | 2.48E-6 | 0.004 |
| GO:0006936~muscle contraction | 16 | 3.87 | MYL4, ACTC1, ACTA2, CALD1, UTRN, VIPR1, GJC1, EDNRB, ACTG2, SSTR2, AGT, MYH11, MYOM1, CACNA1C, HRC, SGCA | 3.69E-6 | 0.006 |
| GO:0050880~regulation of blood vessel size | 10 | 2.42 | CAV2, ACTG2, EDNRB, CAV1, ACTC1, P2RX1, ACTA2, AGT, GUCY1A3, HBB | 3.87E-6 | 0.007 |
| GO:0014706~striated muscle tissue development | 14 | 3.38 | CAV2, ACTC1, CAV1, TBX2, UTRN, RXRG, CACNB2, PTEN, GJC1, PLN, MYH11, RARB, MKL2, AGRN | 5.01E-6 | 0.008 |
| **KEGG_Pathway-Term** | **Count** | **%** | **Genes** | **P value** | **FDR** |
| hsa04270:Vascular smooth muscle contraction | 21 | 5.08 | ADCY3, ADCY4, ADCY1, ROCK1, ACTA2, PPP1R12B, ADCY5, CALD1, ADCY6, MRVI1, PRKG1, KCNMB1, ACTG2, GUCY1A2, PPP1R12A, MYH11, GUCY1A3, GUCY1B3, CACNA1C, MYLK, PLA2G5 | 4.00E-11 | 4.55E-8 |

Enrichment was performed using DAVID web software based on Gene Ontology and KEGG databases. *P* values are indicated without or after FDR correction

Both CAF-S1 and CAF-S4 subsets are the most frequent sub-populations detected in mesenchymal HGSOC, suggesting they categorize distinct patients and could exert different functions. CAF-S1 cells are likely to exert immunosuppressive activities that could account for the poor survival of mesenchymal HGSOC patients. We first observed that the stromal compartment shows a higher number of T lymphocytes (especially regulatory T cells), compared to the epithelium, suggesting that CAF are more prone to attract immune cells than cancer cells. This statement has previously been described in murine models and in several human cancers[38,49–52]. In that sense, FAP$^{Pos}$ cells have been shown to promote immunosuppression by triggering an inflammatory phenotype, further enhancing the recruitment of myeloid-derived suppressor cells (MDSC)[50–53]. Accordingly, we demonstrate here that CAF-S1 fibroblasts exhibit immunosuppressive capacities in vitro acting on CD25$^+$FOXP3$^+$ T lymphocytes at three levels by enhancing their attraction, survival, differentiation, and suppressive activity. CAF-S1 cells have thus functional similarities to MDSC, well-known to display immunomodulatory

functions[54–57]. Human MDSC are commonly described as Lin-CD11b + CD33 + HLADR− cells, with neutrophilic or monocytic phenotype[57]. In contrast, none of the markers characterizing MDSC, such as CD11b$^+$ or CD33$^+$, are detected in CAF-S1 cells (EBI study accession number: EGAS00001002184). Furthermore, we show that the CXCL12 chemokine is a key player involved in attraction of regulatory T cells in human HGSOC, confirming previous studies showing the role of CXCL12 in immunosuppression[37,38]. These effects are independent of CXCL12-proangiogenic function[27,58,59], as we detected them using functional assays in vitro. Accordingly, CXCL12/CXCR4 blockade reduces recruitment of regulatory T lymphocytes in immuno-competent mouse models of cancers[28,29,37,38]. In contrast, we did not observe tumor growth inhibition upon CXCR4 inhibitor (AMD3100) treatment in immuno-deficient patient-derived xenograft (PDX) models of ovarian cancer (Supplementary Fig. 7), suggesting that immunosuppressive functions mediated by both *CXCL12α* and *CXCL12β* isoforms on T lymphocytes could be crucial for ovarian tumor growth. In that sense,

detection of CD3[+] and CD8[+] T lymphocytes in tumor epithelial compartment is correlated with improved patient survival in HGSOC[60,61]. Conversely, regulatory T lymphocytes infiltration predicts shortened patient survival[62–64], suggesting that immunosuppressive environment fostered by regulatory T cells plays a key role in this pathology. Finally, we identified B7H3, CD73, and IL6, as new actors involved in CD25[High]FOXP3[High]

differentiation. These genes are highly expressed by CAF-S1 fibroblasts and their silencing in CAF-S1 significantly reduces the proportion of CD25[High]FOXP3[High] T-lymphocyte population, enriched in regulatory T lymphocytes. These observations could thus pave the way for the use of the already-existing drugs targeting IL6, B7H3 or CD73, in order to enhance anti-tumor immunity in CAF-S1-enriched HGSOC.

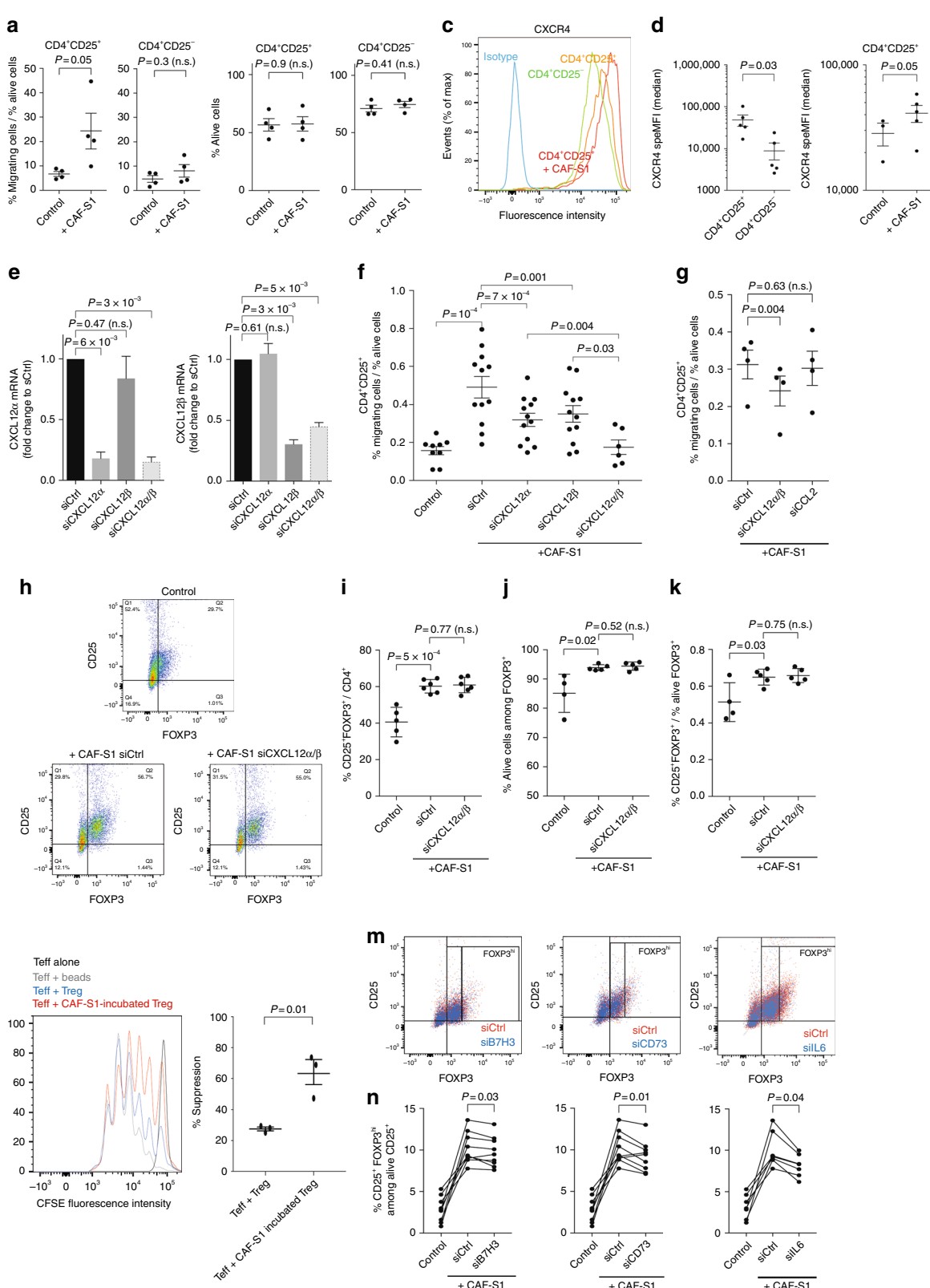

Studies investigating the impact of CXCL12 and CXCR4 protein levels on HGSOC patient survival remain highly controversial[22–24,65,66]. Among the six *CXCL12* isoforms that have been previously described[67,68], *CXCL12α* and *CXCL12β* are the major ones showing ubiquitous expression pattern in human tissues[67,68]. However, the specific regulation and functions of the different CXCL12 isoforms in human cancers remained unknown and have never been investigated in ovarian cancers. We highlight here the specific regulation of the *CXCL12β* isoform in mesenchymal HGSOC, which has never been reported before. Indeed, we demonstrated that the *CXCL12β* isoform strictly accumulates in CAF-S1 fibroblasts through a post-transcriptional regulation by miR-141/200a in CAF-S4 fibroblasts in HGSOC. Interestingly, this regulation has also been observed in Crohn's disease[69]. Moreover, the miR-200 family members are highly relevant in mesenchymal HGSOC, as these tumors have been identified by genes anti-correlated with miR-141/200a[7,13]. The difference in the protein sequences between CXCL12α and CXCL12β isoforms only resides in the presence of four additive amino acids in the C-terminal part of CXCL12β. These four amino acids are known to enhance the interaction of the chemokine with the surrounding proteoglycans and promote its maintenance inside the tissue[68]. Thus, it suggests that high expression of the CXCL12β isoform not only promotes the release of this chemokine in the tumor microenvironment of mesenchymal HGSOC, but also its maintenance in close vicinity of ovarian cancer cells. Taken as a whole, CAF-S1 cells expressing high levels of CXCL12β and CXCL12α isoforms could constitute an immunosuppressive environment in mesenchymal HGSOC. Finally, based on this immunosuppressive environment, we could speculate that HGSOC enriched in CAF-S1 fibroblasts would not benefit from immunotherapies, as it has been shown for CXCL12 in mouse models of pancreatic cancer[37,38]. Indeed, our data suggest that high content in CAF-S1 fibroblasts within tumors could be indicative of their intrinsic resistance to immunotherapies. Evaluating the proportion of CAF-S1 in mesenchymal HGSOC might be relevant for discriminating patients that would be resistant or sensitive to immunotherapies. Thus, targeting CAF-S1 fibroblasts by using CAF-S1-directed drugs could be very promising for improving sensitivity of mesenchymal HGSOC to immunotherapies.

## Methods

**Cohorts of HGSOC patients.** Cohorts of ovarian cancer patients from Institut Curie (107 patients for transcriptomic data and 118 patients with available tumor samples for IHC), AOCS (285 patients), and TCGA (484 patients) have been previously described in refs. [6–8,12]. Main clinical features of these cohorts have also been summarized in Table 1 and its corresponding legend. In total, 107 ovarian tumor samples from the Institut Curie cohort have been considered only if they were enriched in at least in 55% of epithelial cancer cells. RNA has been subsequently extracted and analyzed using Human Genome U133 Plus 2.0 microarray

(Affymetrix), as described in ref. [7]. Microarray data from Institut Curie cohort are freely accessible in the Gene Expression Omibus under the accession number GSE26193. For the TCGA cohort, freely available transcriptomic data (L2 level) obtained using Human Genome U133A microarray (Affymetrix)[8] have been downloaded from the following portal: http://cancergenome.nih.gov/. The microarray data set of the AOCS cohort has been obtained with U133 Plus 2 microarrays (Affymetrix) and is freely accessible under the accession number GSE9899[6]. For performing IHC, samples from the Institut Curie cohort were first selected based on tumor grade, histological subtype, clinical features for considering only HGSOC. Among 139 HGSOC[7,12], only 118 HGSOC with available remaining tumor tissues could be analyzed (Table 1). For FACS analyses, 45 HGSOC samples were collected from the operating room after specimen's macroscopic examination and selection of areas of interest by a pathologist. Patients have been recruited from 2012 to 2016. Phenotypic analyses were performed from these 45 HGSOC samples. For performing RNA seq, CAF subsets were sorted (see below for precise description of the method), and RNA extracted. A total of 13 different CAF-S1 and CAF-S4 samples of good quality for RNA and cDNA was subsequently submitted to RNA sequencing (see below).

The project developed here studying samples from the Institut Curie is based on surgical tumor tissues available after histo-pathological analyses that are not needed for diagnostic purposes. There is no interference with the clinical practice. Analysis of tumor samples was performed according to the relevant national law on the protection of people taking part in the biomedical research. Their referring oncologist informed all patients included in our study that their biological samples could be used for research purposes and patients signed an informed consent of non-opposition. In case of patient refusal, which could be either orally expressed or written, residual tumor samples were not included in our study. The Institutional Review Board and Ethics committee of the Institut Curie Hospital Group approved all analyses realized in this study, as well as the CNIL (Commission Nationale de l'Informatique et des Libertés), the National Commission for Data Processing and Liberties (N° approval: 1674356 delivered on March 30, 2013). Mesenchymal and non-mesenchymal HGSOC molecular subtypes have been defined as in refs.[7,14]. In the retrospective cohort of patients, all samples included in our study have been collected before any treatment. In the prospective cohort, we collected all available tumor biopsies, independently of the treatment prior surgery, as patient information was not systematically accessible at time of surgery. However, we analyzed samples retrospectively and compared CAF contents at time of surgery, with or without neo-adjuvant chemotherapy, and found almost no variation in CAF subset content in the samples collected here (Supplementary Fig. 2d).

**Cell sorting from HGSOC samples and FACS analysis.** Fresh human HGSOC samples were obtained directly from the operating room, after surgical specimen's macroscopic examination and selection of areas of interest for diagnosis by a pathologist. To prepare cell suspensions from HGSOC, tumors were cut into small pieces (around 1 mm$^3$) and submitted to an enzymatic digestion in $CO_2$-independent medium (Gibco #18045-054) supplemented with 5% fetal bovine serum (FBS, PAA #A11-151), with 2 mg/ml collagenase I (Sigma #C0130), 2 mg/ml hyaluronidase (Sigma # H3506) and 25 μg/ml DNase I (Roche #11284932001) during 45 min (min) at 37 °C with agitation (160 rpm). Cells were then filtrated through a 40 μm cell strainer (Fisher Scientific #223635447) and resuspended in PBS+ solution (PBS, Gibco #14190; EDTA 2 mM, Gibco #15575; Human Serum 1 %, BioWest #S4190-100) at a concentration of 5×10$^5$ cells in 50 μl.

For detection of both cell surface (EPCAM, CD45, CD31, FAP, PDGFRβ, CD29, Caveolin) and intracellular (SMA, FSP1) proteins (referred to as intracellular staining performed in Fig. 3a–c) in HGSOC cell suspensions, most of the antibodies (Supplementary Table 1 for references) were purchased already conjugated with fluorescent dyes except the anti-FAP and anti-FSP1 (S100A4) antibodies. The FAP-directed antibody was conjugated with the Zenon Pacific Orange Mouse IgG1 labeling kit (ThermoFisher Scientific #Z25269) before use, and an Alexa Fluor 647-RPE-conjugated goat anti-rabbit IgG secondary antibody

**Fig. 5** CAF-S1 stimulate CD25$^+$FOXP3$^+$ T lymphocytes. **a** Percentage of CD4$^+$CD25$^+$ or CD4$^+$CD25$^-$ lymphocytes migrating towards CAF-S1. Data are shown as mean ± SEM. $n = 4$ independent experiments. P values from Student's t-test. **b** Percentage of alive CD4$^+$CD25$^+$ or CD4$^+$CD25$^-$ T lymphocytes in absence (Control) or presence of CAF-S1. Data are shown as mean ± SEM. $n = 4$. P values from Student's t-test. **c** Density curves showing CXCR4 expression in CD4$^+$CD25$^-$ (green), untreated CD4$^+$CD25$^+$ (orange) or after culture with CAF-S1 (red), compared to control isotype (blue). Cell count is normalized, as percentage of maximal number of cells (% of max). **d** CXCR4 protein levels in CD4$^+$CD25$^-$ or CD4$^+$CD25$^+$ lymphocytes without (Control) or with CAF-S1. Specific MFI are shown as mean ± SEM. $n = 5$. P values from paired ttest. **e** *CXCL12α* and *CXCL12β* mRNA levels after silencing of CXCL12α- or/and CXCL12β in CAF-S1 cells. Data are shown as mean ± SEM of fold change to control. $n = 5$. P values from one sample ttest. **f** Percentage of migrating CD4$^+$CD25$^+$ lymphocytes after CXCL12α/β silencing in CAF-S1 cells. Data are shown as mean ± SEM. $n = 5$. P values from paired ttest. **g** Same as in (**f**) after CXCL12α/β or CCL2 silencing. $n = 3$. **h** Flow cytometry plots showing CD4$^+$CD25$^+$ and FOXP3$^+$ cells in absence (Control) or presence of CAF-S1 transfected with siCtrl or silenced for both CXCL12α and β (siCXCL12α/β). **i, j, k** Percentages of CD25$^+$FOXP3$^+$ among CD4$^+$ cells (**i**), of alive CD25$^+$FOXP3$^+$ lymphocytes (**j**) and of CD25$^+$FOXP3$^+$ lymphocytes relative to alive cells (**k**). Data are shown as mean ± SEM. $n = 4$. P values from Student's t-test. **l** CFSE fluorescence intensities quantifying CD4$^+$ effector T cells (Teff) proliferation. Teff were incubated alone (black curve), with CD3$^+$CD25$^+$ beads (grey), or in presence of CD4$^+$CD25$^{High}$CD127$^{low}$CD45RA$^{low}$ (+Treg) either pre-incubated with CAF-S1 fibroblasts (red) or not (blue). Scatter plot shows percentage of suppression (see Methods). **m** Flow cytometry plots showing CD4$^+$CD25$^+$FOXP3$^{+/-}$ cells without (Control) or with CAF-S1 transfected with siCtrl or silenced for B7H3, CD73 or IL-6. **n** Percentages of alive FOXP3$^{High}$ lymphocytes as in (**m**). P values from paired ttest. $n \geq 6$

(ThermoFisher Scientific #A-20991) was used for FSP1 detection. For intracellular staining, cells were first stained 20 min at room temperature (RT) with a violet LIVE/DEAD marker (ThermoFisher Scientific #L34955) in PBS (Gibco #14190) and then fixed in paraformaldehyde (PFA, Electron Microcopy Sciences #15710) 4% overnight (ON) at 4 °C. After a rapid washing step in PBS+ solution, cells were stained using the pool of primary antibodies (conditions of dilution used for each antibody specified in Supplementary Table 1) 45 min at RT in PBS+ solution with

0.1% Saponin and finally incubated (for FSP1 detection) with an anti-rabbit secondary antibody (ThermoFisher Scientific #A-20991) during 15 min at RT in PBS+ medium with 0.1% Saponin. In all experiments, corresponding isotype control antibody was used for each CAF marker antibody (listed in Supplementary Table 1). For surface staining, detecting proteins located at the plasma membrane and compatible with RNA extraction (used in Fig. 5a), cell suspensions were stained immediately after dissociation of HGSOC samples and filtration by using a

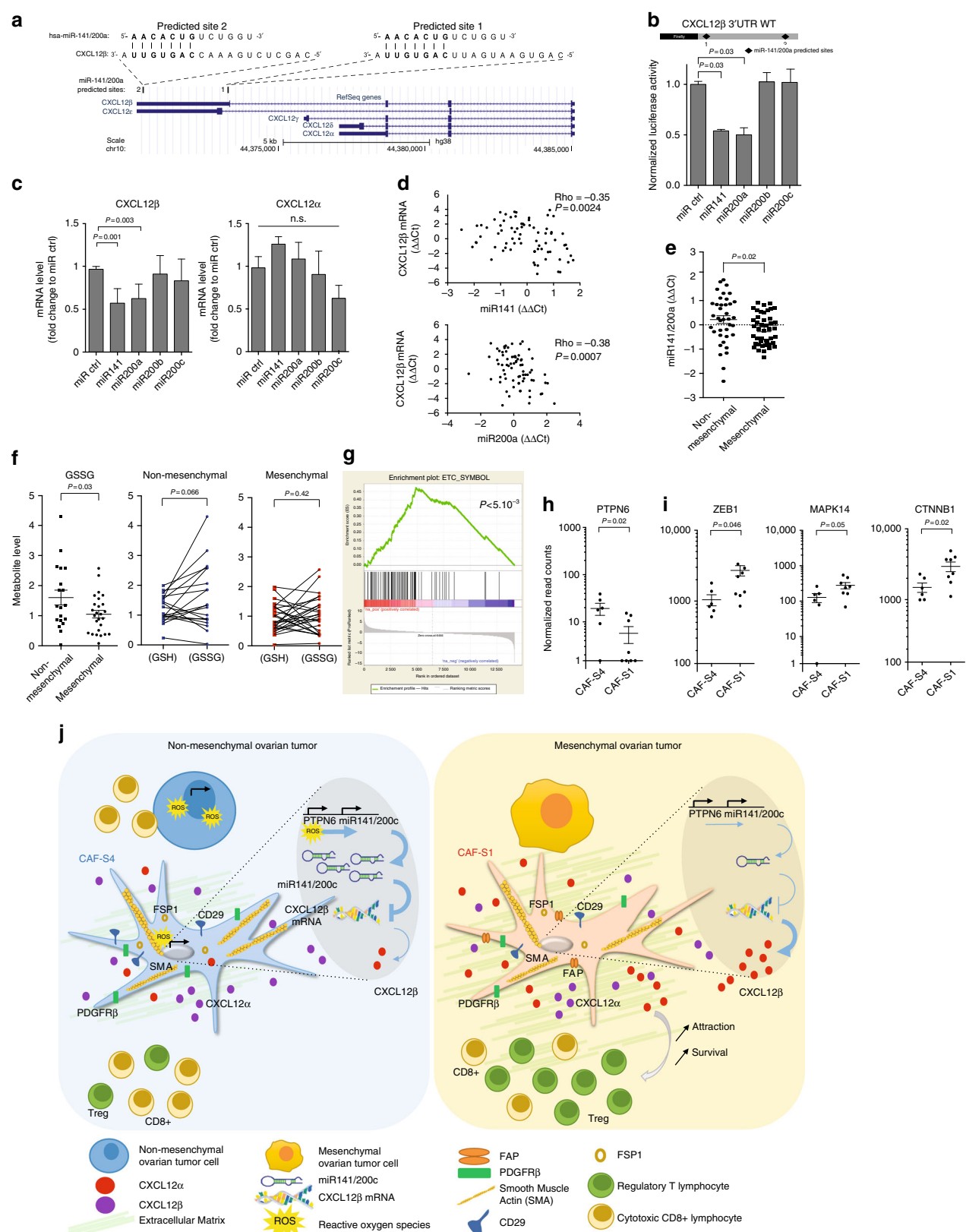

pool of antibodies recognizing EPCAM, CD45, CD31, FAP, PDGFRβ, and CD29 (same as those used for intracellular staining), during 15 min at RT in PBS+ solution. DAPI (3 μm) (ThermoFisher Scientific #D1306) was added just before flow cytometry analysis.

In the two conditions (surface and intracellular staining), cells were analyzed on the LSRFORTESSA analyzer (BD Biosciences) for flow cytometry analysis. At least $5\times10^5$ events were recorded. Compensations were performed using single staining on anti-mouse IgG and negative control beads (BD Biosciences #552843) for each antibody and on ArC reactive beads (Molecular Probes #A10346) for Live/Dead staining. Data analysis was performed using FlowJo version X. Cells were first gated based on forward (FSC-A) and side (SSC-A) scatters (measuring cell size and granulosity, respectively) to exclude debris. Single cells were next selected based on SSC-A versus SSC-W parameters. Dead cells were excluded based on their positive staining for Live/Dead (fixed conditions) or DAPI (surface staining). Cells were then gated on EPCAM$^-$, CD45$^-$, CD31$^-$ cells, for excluding epithelial cells (EPCAM$^+$), hematopoietic cells (CD45$^+$) and endothelial cells (CD31$^+$). DAPI$^-$, EPCAM$^-$, CD45$^-$, CD31$^-$ cells were next examined using the six CAF markers including CD29, FAP, SMA, FSP1, PDGFRβ, and Caveolin. Sorting of CAF subsets after surface staining was performed on FACSARIA (BD Biosciences). SPADE algorithm was performed using Cytobank (Cytobank Inc., Mountain View, CA, USA). SPADE clustering was performed using 300 nodes to generate unified trees based on the expression of the six CAF markers. CAF subset populations were manually annotated.

**Gene expression profiling of CAF subsets.** CAF-S1 and CAF-S4 subsets from fresh HGSOC samples were collected after FACS in RNAse-free tubes (Thermo-Fisher Scientific #AM12450), using the same strategy as described above (cell surface staining compatible with RNA extraction). At least 100 cells and up to 4000 cells were collected for each CAF subset from fresh HGSOC. Total RNA from CAF-S1 and CAF-S4 were extracted with Single Cell RNA Purification kit (Norgen Biotek #51800). RNA integrity and quality was verified using Agilent RNA 6000 Pico Kit (Agilent, #5067-1513) and apparatus. cDNA libraries were synthesized using SMARTer Ultra Low input RNA kit (Clonetech #634820 #634823 #634826 #634828, and #634830). cDNA quality was checked on Agilent 2100 bioanalyser using Agilent High Sensitivity DNA kit (Agilent #5067-4626). cDNA library was prepared using Nextera XT preparation kit (Illumina #FC-131-10). Samples were sequenced on a rapid run flow cell of HiSeq 2500 (Illumina) with an average sequencing depth of 30 millions of paired-end reads. Length of the reads was 100 bp. Reads were mapped on reference human genome (hg19/GRCh37 from UCSC genome release) using Tophat_2.0.6 algorithm with the following parameters: global alignment, no mismatch in seed alignment (of size 22), three mismatches in read length. Quality control was performed using FastQC software and duplicates were removed using Samtools rmdup. Quantifications of expression at both gene and transcript levels were performed using HTSeq-count and featureCouts (implemented in Bioconductor R package Rsubread). Only genes with one read in at least 5% of all samples were kept for further analyses. Normalization was performed using method implemented in DESeq2 R package. Analysis strategy includes unsupervised analysis such as PCA and HC, as well as differential expression analysis (done with DESeq2 bioconductor package). Biological interpretations of the genes (or transcripts) identified were assessed by computational functional analyses based on several bioinformatics resources (Gene Ontology, KEGG, Reactome, Ingenuity Pathway Analysis). Accession number: RNAseq data from CAF-S1 and CAF-S4 fibroblast sorted from HGSOC samples have been submitted at the European Genome-Phenome Archive (EGA) at EMBL-EBI (EBI study accession number: EGAS00001002184).

**Immunohistochemistry staining on HGSOC sections.** A total of 118 HGSOC samples have been selected by pathologists, as representative of the pathology. Consecutive sections of paraffin-embedded HGSOC tissues (3 μm) were stained using streptavidin-peroxidase protocol (Vectastain ABC kit; Vector Labs #PK-

6101) on the Autostainer Labvision (Thermoscientific). In brief, paraffin-embedded sections were incubated with specific antibodies recognizing CXCL12, CXCR4, CAF (CD29, FAP, PDGFRβ, SMA, FSP1) and T-lymphocytes markers (CD3, CD8, FOXP3) (Supplementary Table 1 for antibody references and dilutions used for immunohistochemistry (IHC)) in PBS solution containing 0.05% Tween 20, for 1 h, following either unmasking in Tris/EDTA buffer, pH = 9 (Dako #S2367) or citrate buffer, pH = 6 (Dako #S2369) (depending of the primary antibody, Supplementary Table 1) for 20 min at 97 °C. For evaluation of blood vessel density, the anti-CD31 (1:100, Dako #M0823) was used. For quantification of each CAF marker, the whole section was considered and evaluated by two independent researchers. Epithelial to stromal compartments were first delineated using E-Cadherin and SMA staining, respectively. For staining of each CAF marker, histological score (Hscore) was given as a function of percentage of stained fibroblasts multiplied by staining intensity (ranging from 0 to 4). For quantifications of T lymphocytes, at least five areas of 0.4 mm$^2$ per tumor were evaluated. We counted the number of CD3$^+$, CD8$^+$ or FOXP3$^+$ T lymphocytes in each compartment (considering the epithelial and stromal compartments separately), and divided by the total area of the section.

**Triple-immunofluorescence staining on HGSOC sections.** Sections of paraffin-embedded HGSOC tissues (3 μm) were consecutively treated with protein block buffer (Dako, #S2369) during 10 min, primary antibody during 1 h and secondary antibody (1/600) during 30 min in the following order for primary antibodies: anti-FAP antibody, anti-SMA antibody, and anti-CD29 antibody (see Supplementary Table 1 for references and conditions of dilution used for each antibody). Sections were then stained with 3 μM DAPI for 15 min at 37 °C and mounted using Vectashield mounting media (Vector Laboratories, #H-1000). Images were acquired at ×20 on ApoTome microscope (Zeiss).

**Decision tree algorithm predicting CAF subset identity.** CAF identity was determined by using an algorithm developed by the team, which takes as input histological scores of CAF markers. In a first step, the thresholds (quartiles and median) and the order of decisions were established using FACS data. These FACS data included expression of each CAF marker in the four CAF (CAF-S1,2,3,4) cellular subpopulations, sorted from fresh HGSOC samples (prospective cohort of HGSOC patients, N = 22). Threshold values were then transposed to IHC data, using a learning set of tumors containing both non-activated and activated CAF (Retrospective cohort, N = 60).

**Maps of CAF subsets.** IHC staining from consecutive sections were scanned on Philips Ultra Fast Scanner. ×10 images of each of the five CAF markers (CD29, FAP, PDGFRβ, SMA, FSP1) and of CD3 staining from the same tumor areas in representative tumors were further analyzed. Images were first aligned by applying an elastic transformation using a Fiji software plugin (bUnwarpJ). The plugin uses landmarks, defined by the user, and haematoxilin and eosin (H&E) staining from the different sections to compute the best correlation between images. The software next applies an elastic transformation to all images to align tissue structures at cellular level. Epithelial cells were masked to facilitate stroma visualization. Images were next divided into tiles (square of 225 μm$^2$ defined as the approximate size of a cell), allowing identification of each cell by a tile. Each tile was then annotated according to specific position in the section. After color deconvolution, intensity of DAB staining for each tile and for each marker was measured by densitometry analysis (ImageJ software). CAF identity per tile was determined by using an algorithm developed by the team (see paragraph above), which takes as input DAB intensities of CAF markers measured within each tile. Epithelial tumor cells were represented in black to better visualize the stromal compartment and each tile was colored with CAF-S1 in red, CAF-S2 in orange, CAF-S3 in green and CAF-S4 in blue. Finally, the distance between CAF subsets and epithelial cells has been established using an algorithm developed in the lab and implemented using R

**Fig. 6** *CXCL12β* mRNA is targeted by miR-141/200a in CAF-S4. **a** Schema (from https://genome.ucsc.edu) of CXCL12 human genomic locus. Two predicted miR141/200a sites are shown, site 1 specific of *CXCL12β* and site 2 also detected in *CXCL12ε*. **b** Normalized luciferase activity of CXCL12β-3'UTR-luciferase reporter construct after co-transfection with miR200s. Values are fold changes of Firefly/Renilla activity ratio (normalized to control) ± SEM. n = 2. P values from Student's t-test. **c** *CXCL12α* and *CXCL12β* mRNA levels in miR-200-overexpressing CAF-S1. Data are mean of fold change ± SEM. n = 3. P values from Student's t-test. **d** Correlation plots between *CXCL12β* mRNA and miR-141 or miR-200a levels. P values from Spearman's test. **e** miR-141/miR-200a levels in non-mesenchymal (N = 38) versus mesenchymal (N = 45) HGSOC (Institut Curie cohort). Normalized cycle thresholds are centered to the mean (ΔΔCt). P value from Student's t-test. **f** Reduced (GSH) and oxidized (GSSG) glutathione levels evaluated by mass spectrometry in non-mesenchymal (N = 19) and mesenchymal (N = 25) HGSOC. P value from Student's t-test (non-mesenchymal/mesenchymal) and paired ttest (GSH/GSSG). **g** Significant enrichment of electron transport chain (ETC) gene signature in CAF-S4. P refers to false discovery rate q-value. **h, i** *PTPN6* (**h**) and *ZEB1, MAPK14, CTNNB1* (miR-141/200a-target genes[7,35,36]) (**i**) mRNA levels in CAF-S1 and CAF-S4. Data are mean ± SEM. P values from Student's t-test. **j** Model, mesenchymal HGSOC[7,13] accumulate a dense stroma enriched in CAF-S1 fibroblasts (Right). CAF-S1, characterized by expression of FAP, SMA and PDGFRβ, promotes attraction of regulatory T cells through CXCL12β and CXCL12α. CAF-S4 fibroblasts accumulate in non-mesenchymal HGSOC (Left), characterized by genes correlated with miR-200[7,13], involved in oxidative stress response. Accordingly, non-mesenchymal HGSOC suffer from a chronic oxidative stress. CAF-S4 fibroblasts exhibit lower levels of *CXCL12β* mRNA than CAF-S1, and thus show reduced attraction of regulatory T lymphocytes. In contrast to *CXCL12α*, *CXCL12β* is targeted by miR-141/200a that accumulate in CAF-S4-enriched non-mesenchymal HGSOC

software. In brief, each tile of the images was annotated according to a specific position in the section, using Cartesian coordinate system in two dimensions ($x$ and $y$ coordinates). For each tile (annotated as a CAF subset based on CAF marker intensities), we calculated the distance with the closest epithelial cell (in a maximum area of five successive tiles in $x$ and $y$) using Euclidean distance, as followed:

$$d(x, y) = \sqrt{(x\mathrm{epith} - x\mathrm{CAF})^2 + (y\mathrm{epith} - y\mathrm{CAF})^2}.$$

Distance was finally represented as a heatmap with color range from red (minimum distance) to yellow (maximum distance). To quantify CD3 co-localization with CAF subsets, aligned CD3 image was analyzed with the corresponding CAF subset map, generated by alignment of the stromal markers, as described above. The number of CD3 at the surface of each CAF subset was next evaluated in HGSOC sections.

**Primary ovarian CAF culture.** Fresh human tumor samples from the operating room were cut into pieces of approximately 5 mm$^3$ and put on petri dishes in DMEM (Gibco #11965092) supplemented with 10% FBS (PAA #A11-151), penicillin (100 U ml$^{-1}$) and streptomycin (100 µg ml$^{-1}$) (Gibco #15140122). After 2 −3 weeks of incubation at 37 °C in 3% O$_2$, fibroblast-like cells were the only ones which grew and could be plated in a new plastic dish. This step corresponded to the first passage. To validate their fibroblast identity, cells were checked for their negative expression of EPCAM, CD45, and CD31 and further characterized for their expression of CAF markers by flow cytometry analysis, using the pool of antibodies described above (# Cell sorting from HGSOC samples and FACS analysis). For functional assays, cells were maintained during a maximum of ten passages, for avoiding them entering into senescence. At least ten different CAF-S1 primary cell lines, isolated from HGSOC, were analyzed and used for functional experiments.

**qRT-PCR from HGSOC and cell lines.** For gene expression analysis, total RNA isolation was performed using miRNEasy kit (Qiagen #217004) according to the manufacturer's instructions. RNA concentrations were determined with a NanoDrp apparatus (NaNodrop Technologies, Inc.). One microgram of total RNA per sample was reverse transcribed using an iScript Reverse Transcription Kit (Bio-Rad #1708840). qRT-PCR was next performed using Power SYBR Green PCR Master Mix (Applied Biosystems #4367659) on a Chromo4 System (Bio-Rad) with primers at 300 nM final concentration. CXCL12 primers were designed using the PrimerQuest tool (Integrated DNA Technologies) to span two exons. Specificity of primers used for detecting each CXCL12 isoform was checked by sequencing the PCR product after qPCR. We selected primer sets with an efficiency of amplification between 95 and 100%. Data were analyzed using an Opticon Monitor (Bio-Rad) and normalized to GAPDH mRNA. The sequences of primers used were the following: CXCL12α-forward: 5′-TCTCAACACTCCAAACTGTGCCCT-3′; CXCL12α-reverse: 3′-TGCCCTTTCATCTCTCACAAGGT-5′; CXCL12β-forward: 5′-AACAGACAAGTGTGCATTGACCCG-3′; CXCL12β-reverse: 3′-TAA-CACTGGTTTCAGAGCTGGGCT-5′; CXCL12γ-forward: 5′-AACAGA-CAAGTGTGCATTGACCCG-3′; CXCL12γ-reverse: 3′-TGGGCAGCCTTTCTCTTCCTTGT-5′; GAPDH-forward: 5′-GAAGGT-GAAGGTCGGAGTC-3′; GAPDH-reverse 3′-GAAGATGGTGATGGGATTTC-5′.

TaqMan qRT–PCR assay was used for detection of miRNAs. Reagents, primers, and probes were obtained from Applied Biosystems (#442795). RT reactions and real-time qPCR were performed according to the manufacturer's protocols from 50 ng of RNA per sample. Primers and probes are specific for each miRNA and are designed by the manufacturer. U6 snRNA and miR-16 were used as loading controls as described in ref. [13]. qPCR reactions were performed in a Chromo4 apparatus (Bio-Rad).

**Silencing of CXCL12 and transient transfection.** For the short interfering RNA (siRNA) experiment, 1.4×10$^5$ CAF-S1 cells were plated in six-well plates. After 24 h, cells were transfected with 20 nM of non-targeting siRNA (siCtrl, Thermo-scientific #D-001810-01-20), CXCL12α-targeting siRNA (siCXCL12α#1, target sequence: 5′-UAAGCUGCAAUAUCAUACCUU-3′; and siCXCL12α#2, target sequence: 5′-UAAGCCACCACCUGACUGUUU-3′) or CXCL12β-targeting siRNA (siCXCL12β#1, target sequence: 5′- UCUGACCCUCUCACAUCUU-GAACUU-3′; and siCXCL12β#2, target sequence: 5′-GGCAAGUA-CAAUAAUGGCCUU-3′) using 4 µl of Dharmafect 1 transfection reagent in 2 ml final volume according to the manufacturer's instructions (Thermoscientific #T-2001-02). For miRNA overexpression in CAF-S1, cells were transiently transfected with 20 nM of miRIDIAN miRNA Mimics (Dharmacon; universal negative control: #CN-002000-01-05; miR-141: #C-300608-03-0005; miR-200a: #C-300651- 05-0005; miR-200b: #C-300582-07-0005; miR-200c: #C-300646-05-0005) using 4 µl of Dharmafect 1 in 2 ml final volume (six-well plates).

**CXCL12β expression profile among different cell types.** We analyzed CXCL12β expression levels among different primary cell types by analyzing publicly available database on Gene Expression Omnibus resources under the accession number GSE49910. Data have been generated from Affymetrix U133plus2.0 array. To have access to CXCL12α and CXCL12β mRNA levels, we considered both 203666_at

and 209687_at probesets. We calculated the average of expression of the two probesets in the different primary cell types analyzed. They are included in the GSE49910 data set, both control, and treated, infected or stimulated cells. We have considered in our analysis, only control primary cells, devoid of any treatment.

**CXCL12 in situ hybridization.** RNAscope® Technology is a commercially available cutting-edge in situ hybridization (ISH) technology that utilizes a branched or "tree" in situ method to obtain ultrasensitive, single transcript detection. Sections of formaldehyde-fixed paraffin-embedded HGSOC tissues (3 µm) were processed following manufacturer description for RNAscope 2.5 assay (ACD, Advanced Cell Diagnostic). Briefly, hydrogen peroxide, target retrieval, and protease reagents (Pretreatment reagents #322300 and #322000) were applied on deparaffined sections at room temperature. Hybridization and amplification steps (RNAscope detection reagents—RED, #322360) were then performed at 40 °C in HybEZ hybridization system (#310010) as described by the manufacturer, using RNAscope CXCL12 probe (ACD, #425251).

**Isolation of CD4$^+$CD25$^+$ cells, transwell assay, and co-culture.** In order to isolate CD4+ CD25+ cells, healthy donor human blood samples were obtained from "Etablissement Français du Sang", Paris, Saint-Antoine Crozatier blood bank through an approved convention with the Institut Curie (Paris, France). PBMC were isolated from blood using Lymphoprep procedure (Stemcell #7861). PBMC were isolated from healthy-patient blood using Lymphoprep procedure (Stemcell #7861). 50×10$^7$ PBMC were used to isolate CD4$^+$CD25$^+$ T lymphocytes in order to enrich the population into regulatory T lymphocytes by using MicroBeads CD4 $^+$CD25$^+$ Regulatory TCell Isolation Kit (Miltenyi Biotec #130-091-041) and high gradient magnetic cell separation columns (MACS) (LS columns, Miltenyi Biotec #130-042-401) according to the manufacturer's instructions.

Five micrometers transwell permeable supports (Corning #CLS3388-2EA) were used for Transwell assays. 150,000 CD4$^+$CD25$^+$ T lymphocytes were placed on the upper part of the Transwell device and incubated in 50 µl of DMEM (Gibco #11965092) supplemented with 1% FBS. The lower chamber either contained 200 µl of DMEM supplemented with 1% FBS alone (control condition) or was plated with 20,000 primary HGSOC CAF-S1 fibroblasts. The experiment was stopped after 14 h of incubation at 37 °C in 3% O$_2$. CD4$^+$CD25$^+$ T lymphocytes that remained in the upper chamber and cells that migrated to the lower chamber were collected separately, incubated with 0.5 µl of 10 µm carboxylate beads (Polyscience #18133) and DAPI solution (3 µM) and analyzed on LSRII cell analyzer (BD Biosciences). Based on FSC and SSC parameters, T lymphocytes and beads were gated separately and T-lymphocyte count was normalized with bead count to evaluate the exact number of cells in each well. Results are shown as a ratio of percentage of T lymphocytes in the lower chamber among the total number of cells, normalized by percentage of live cells.

To perform co-culture experiments, 600,000 CD4$^+$CD25$^+$ T lymphocytes were placed into 12-well plates, previously plated with 75,000 primary HGSOC CAF-S1 fibroblasts in 1 ml of DMEM supplemented with 1% FBS or with 1 ml of DMEM supplemented with 1% FBS only (control condition). After 24 h of incubation at 37 °C in 5% CO$_2$ and 1.5% O$_2$, CD4$^+$CD25$^+$ T lymphocytes were collected. Cells were stained for 20 min at RT with a violet LIVE/DEAD marker (ThermoFisher Scientific #L34955) in PBS. After a rapid wash with PBS+ solution, T lymphocytes were stained during 15 min at RT with a pool of fluorescent-conjugated primary antibodies recognizing CD45, CD3, CD4, CD25, and CXCR4 proteins (see Supplementary Table 1 for references and conditions of dilution used for each antibody). Cells were then fixed and permeabilized in the appropriate buffer from the FOXP3 staining buffer set (Ebioscience #00-5523-00) ON at 4 °C. After a wash in a second buffer from the set, cells were stained with a anti-human FITC-FOXP3 antibody 30 min at RT or incubated with the corresponding Rat FITC-IgG2a isotype control (Supplementary Table 1). Cell analysis was performed on LSRFORTESSA (BD Biosciences). At least 5×10$^5$ events were recorded. Compensations were performed using single staining on anti-mouse IgG and negative control beads (BD Biosciences #552843) for mouse antibodies, on AbC Total compensation beads (Molecular Probes #A10513) for rat antibody and on ArC reactive beads (Molecular Probes #A10346) for Live/Dead staining. Data analysis was performed using FlowJo version X. Cells were first gated based on forward (FSC-A) and side (SSC-A) scatters (measuring cell size and granulosity, respectively) to exclude debris. Single cells were next selected based on SSC-A versus SSC-W parameters. Dead cells were excluded based on their positive staining for Live/Dead. Cells were then gated on CD45$^+$, CD3$^+$, CD4$^+$ cells and next examined for their CD25, FOXP3, and CXCR4 expressions (see Supplementary Table 1 for references of each antibody and Supplementary Fig. 6e, for detailed description of the gating strategy).

The Treg suppressive assay was adapted from ref. [70], Benoit Salomon's lab protocol available on protocol exchange. CD4$^+$CD25$^+$ were isolated from healthy donor PBMCs as described earlier. The negative fraction (CD4$^+$CD25$^-$ cells containing T effector cells) was also recovered and kept overnight at 4 °C in RPMI $^-$1640 medium with 2 mM L-glutamine (Hyclone #SH30027.01) supplemented with 10% heat-inactivated FBS (PAA #A11-151), penicillin (100 U ml$^{-1}$) and streptomycin (100 µg ml$^{-1}$) (Gibco #15140122). CD4$^+$CD25$^+$ cells were then stained with a pool of fluorescent-conjugated primary antibodies recognizing CD4, CD25, CD127, CD25, and CD45RA proteins (see Supplementary Table 1 for

references and conditions of dilution used for each antibody) and with DAPI. After a washing step, DAPI⁻CD4⁺CD25^high CD127⁻CD45RA^low cells (regulatory T cells, Treg) were sorted on FACSARIA (BD Biosciences) and pre-incubated overnight at 37 °C, 5% CO$_2$, 1.5% O$_2$ into 24-well plates, previously plated with 50,000 primary HGSOC CAF-S1 fibroblasts in 1 ml of DMEM supplemented with 1% FBS or with 1 ml of DMEM supplemented with 1% FBS only (control condition). After 16 h of incubation, Treg cells were recovered, counted, and re-suspended in RPMI-1640 medium with 2 mm L-glutamine, 10% heat-inactivated FBS, penicillin, and streptomycin. Previously isolated CD4⁺CD25⁻ cells were stained 15 min at 37 °C with 1 μM CellTrace™ CFSE Cell Proliferation dye (ThermoFisher #C34554) at 1×10$^7$ cells per ml in PBS. Suppression assay was performed in U-bottom 96-well plates (Falcon, #353077) during 4 days at 37 °C, 5% CO$_2$, 20% O$_2$: CFSE-stained Teff cells (1×10$^4$ cells/well) were incubated with Treg cells (Treg: Teff ratio = 1:1) pre-incubated with CAF-S1 or with medium only (control condition) and with anti-CD3/CD28 beads (Gibco, #11131D, 10$^3$ beads/well). Wells with CSFE-stained Teff cells alone and wells with CSFE-stained Teff cells, in presence of anti-CD3/CD28 beads but no Treg cells, were used as a negative and positive control of proliferation, respectively. After 4 days, cells were stained with anti-CD4 antibody (see Supplementary Table 1 for references and conditions of dilution) and DAPI and analyzed on LSRFORTESSA analyzer (BD Biosciences). FITC fluorescence (corresponding to CFSE dye) was measured on DAPI⁻CD4⁺ cells. The percentage of suppression was defined as: ((Log2($y$) of Teff alone − Log2($y$) of Teff + Treg)/Log2($y$) of Teff alone)×100, where $y$ is the mean fluorescence intensity (MFI) of CFSE on the whole population divided by the MFI of CFSE of non-proliferating cells.

**UTR luciferase vectors and luciferase reporter assays.** Full-length human CXCL12β 3′-UTR luciferase reporter plasmid was bought from Switchgear Genomics (Menlo Park, USA #S813893). Twenty-four hours prior to transfection, 20,000 293T cells were plated in 96-well without antibiotics. Transient transfection was performed by mixing 0.15 μl of DharmaFECT Duo Reagent (Dharmacon #T-2001-02) with 100 ng of 3′-UTR reporter plasmid and a respective final concentration of 10 nM of miRIDIAN microRNA Mimics (Dharmacon #C-300608-03-0002 #C-300651-05- 0002 #C-300582-07- 0002 #C-300646-05- 0002 #CN-002000-01- 05). As an internal control, 10 ng phRL-TK vector (Promega) were co-transfected and Renilla luciferase activity was used for normalization. Twenty-four hours after transfection, Dual Luciferase Reporter Assay (Promega #E1910) was performed and luminescence was recorded with a Fluostar Optima microplate reader (BMG Labtech).

**Patient-derived xenograft experiment.** Patient-derived xenograft (PDX) models of mesenchymal HGSOC tumors were established at the Institut Curie (Paris, France) with patient consent according to the relevant national law on the protection of people taking part in biomedical research. Tumor fragments of 30−60 mm$^3$ were grafted into the interscapular fat pad of 6-week-old female Swiss nude mice, under avertin anesthesia. When tumors reached a volume of 40−120 mm$^3$, approximately 1 month after grafting, mice were randomly assigned to each group, and the treatments were started. At least nine PDX mice per group were either untreated (control group) or treated by intra-peritoneal injection five times a week for 25 days with CXCR4 inhibitors either AMD3100 (Sigma-Aldrich, #A5602, Saint-Quentin Fallavier, France) or TN14003 (Synthesized by Bachem Company, Budendorf, Switzerland) at a dose of 7.5 mg/kg. Tumor size was evaluated by measuring two perpendicular diameters of tumors with a caliper, twice a week until they reach ethical size. Individual tumor volumes were calculated as $(V) = a \times b^2/2$, with $a$ being the major and $b$ the minor diameter. For each tumor, the tumor volume at day $n$ ($V_n$) was reported as the initial volume before inclusion in the experiment ($V_0$) and expressed as relative tumor volume (RTV) according to the following formula: RTV = $V_n/V_0$. Mean and standard error of the mean (SEM) of RTV in the same treatment group were calculated, and growth curves were established as a function of time. All protocols involving mice and animal housing were in accordance with institutional guidelines as proposed by the French Ethics Committee and have been approved (agreement number: CEEA-IC #115: 2013-06).

**Statistical analysis.** To evaluate the prognostic value of CXCL12 based on transcriptomic data from TCGA, AOCS and Institut Curie cohorts of patients, we computed Log-Rank test using successive iterations (as shown in Supplementary Fig. 3) to find the optimal sample size threshold that maximally discriminate the "Low" and the "High" subgroups of patients. The cut-off value was thus defined as the one that maximally discriminates the two patient subsets in each cohort. When these cut-off values include the median value, the median value was used. Survival analyses were carried out using Kaplan−Meier method and $P$ values were computed by Log-Rank test using survival R package. Data shown in this paper are generally represented as mean ± SEM (unless otherwise specified) from at least three independent experiments.

All along the study, the graphical representation of the data and statistical analyses were done using R environment (https://cran.r-project.org) and GraphPad Prism software. Barplots or scatter plots are represented with mean ± standard error of the mean (SEM) from at least three independent experiments, as indicated in the figure legends. Statistical tests used are in agreement with data distribution: Normality was first checked using the Shapiro–Wilk test and parametric or non-parametric two-tailed test was applied according to normality respect. When normal distribution was observed, equality of variances was then tested using Bartlett's test. If variances between groups were similar, Student's $t$-test was performed. Statistical tests have been indicated in the legends. Differences were considered to be statistically significant at values of $P \leq 0.05$ (bilateral tests). Spearman's correlation test was used to evaluate the correlation coefficient between two parameters.

**Data availability**. RNAseq data from CAF-S1 and CAF-S4 fibroblast sorted from HGSOC samples have been deposited at the European Genome-Phenome Archive (EGA) at EMBL-EBI under the accession number: EGAS00001002184. The results shown here are in part based upon data generated by the TCGA Research Network and available in a public repository from the http://cancergenome.nih.gov/ website. The authors declare that all the other data supporting the findings of this study are available within the article and its supplementary information files and from the corresponding author upon reasonable request.

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

## Acknowledgements

We are grateful to Fariba Nemati and Lisa Maranghoni from the preclinical investigation laboratory at the Institut Curie for expert technical assistance in establishing PDX mouse models, Renaud Leclere and A. Nicolas from the experimental pathology platform at the Institut Curie for help and advice, O. Renaud and the PICT platform at the Institut Curie for technical assistance with microscopy and O. Mariani from CRB Institut Curie for her help in preparing the mRNA and protein samples from patient tumors. The results presented here are in part based upon data generated by the TCGA Research Network: http://cancergenome.nih.gov/. We are grateful to all members of the animal facilities of Institut Curie for their helpful expertise. A.-M.G. was supported by funding from the Cancéropôle Ile-de-France and Ligue Nationale Contre le Cancer. Y.K. was supported by the Institut National du Cancer (INCa) and the Fondation pour la Recherche Medicale (FRM). M.C. was supported by the SiRIC-Curie program (INCa-DGOS-4654). The experimental work was supported by grants from the Institut National de la Santé et de la Recherche Médicale (Inserm), the Institut Curie, in particular the PIC TME and the PIC3i CAFi, the Ligue Nationale Contre le Cancer (Labelisation), the Fondation ARC and INCa (2011-1-PLBIO-12-IC-1 and 2015-1-RT-04-ICR-1). We are very grateful to our funders for having provided their support over the past years.

## Author contributions

F.M.-G. and A.-M.G. participated in the conception and design of the experiments. A.-M. G., A.S.-D., P.S., M.C., F.P., A.C., I.M., and V.M. performed the experiments. G.G. performed the metabolic measurements. Y.K. performed bioinformatic and statistical analyses of the data. C.B. and A.V.-S. provided human samples analyzed in the study. F.

M.-G. supervised the entire project and wrote the paper with A.-M.G. and suggestions from all authors.

## Additional information

**Competing interests:** The authors declare no competing interests.

