## [Peer Review File · Nature Communications]

Reviewers' comments:

Reviewer #1 (Remarks to the Author):

This is an ambitious and wide-ranging study which makes a number of novel and interesting claims. The major claims include identification of novel multi-marker-defined clinically relevant HGSOE CAF subsets, of which one (CAF-S1) displays immune-suppressive actions, which at the molecular level are linked to an expression of CXCL12beta, which is suppressed in other CAF subsets through expression of miR200.

These are all interesting findings of biological and clinical potential. However, with reference to recent "houses of brick; mansions of straw" Kaelin comment in Nature (Vol. 545, page 387) authors and editors are encouraged to select a set of these claims for substantial validation in the preparation of a revised and more focused study.

Comments below are focusing on the first two of the claims identified above, which are estimated to be of most interest.

Major points:

The novelty in the approach of a six-marker-based FACS characterization of clinically derived CAFs is recognized.

A. The subset classification implies the existence of specific marker combinations (e.g. S2 cells with FSPmed/high and low SMA, FAP and CD29) and the absence of cells with other possible combinations (e.g. "only-caveolin-high"). These findings should be validated by selected sets of double- or triple IF analyses of clinical samples.

B. The CAF classification in 1D-G and K is done on an individual cell basis and defining features are summarized in 1E-F. However, the classification of cases in H-J appears to be derived from a score established based on the marker status determined on a case-basis, rather than a cell-basis. The study would be improved with regard to stringency if the "case-classification" were derived from analyses of cell-based maps as those shown in Fig. 1 K.

C. The S1 subset is mostly defined by its FAP-high status. This should be further high-lighted.

D. The mapping approach of Fig. 1K can be made very much more informative by adding information about spatial distribution of the CAF subsets with regard to vicinity to malignant cells, vasculature or immune cell clusters.

E. The identification of associations between the CAF-S1 subset and immune profile (Fig. 2) should acknowledge earlier literature which has implied immune-suppressive roles of FAP-positive fibroblasts. Authors should more explicitly define how the findings of the present study go beyond earlier literature on immune-suppressive effects of FAP+ CAFs, including refs 35, 48 and e.g. Yang, Lin et al., Can Res, 2016 (not cited).

F. The interesting S1/immune cell relationships discussed in Fig. 2 should be extended to analyses where spatial relationships between CAF and immune cell subsets are analysed; are there e.g. any evidence of restriction of a certain immune cell subsets in areas dominated by a particular CAF subset?

G. The data should allow analyses of possible correlations between outcome and abundance, or spatial distribution, of the CAF subsets. Such analyses appear more relevant and novel than the survival association reported for CXCL12beta (Fig. 3G).

Reviewer #2 (Remarks to the Author):

This is an original and carefully conducted study of fibroblasts in high grade serous ovarian cancer, HGSC that contributes to our understanding of the role of fibroblasts in the tumor microenvironment and also extends our knowledge of the role of CXCR4 and CXCL12 isoforms in this disease. It is commendable that all work has been conducted on primary cells isolated from patients or human peripheral blood T cells.

1. Were all the biopsies obtained from the patients prior to treatment? In some cases patients with HGSC receive neo-adjuvant chemotherapy before surgery and this may affect fibroblast activation.
2. Can the authors justify the statistical methods used in some of the data analysis? For instance, they use of parametric tests on data that look non-normal, and T-tests for multiple comparisons.
3. As chemokines bind to ECM proteins and localising them in tissue is not always easy using IHC, the use of in situ hybridisation e.g. with RNAscope to detect CXCL12 mRNA would be a useful addition to this paper.

Reviewer #3 (Remarks to the Author):

This manuscript by Givel and colleagues assesses the role of CAF in HGSC immunopathogenesis. The manuscript asserts that a specific CAF-S1 population attracts immunosuppressive Tregs through CXCL12/CXCR4 and that this is regulated by miR200a in the CAF. Much interesting data is presented and the work is relevant and topical. However, there are loose mechanistic ends that reduce the potential impact of the work in its current form. The biggest issues are showing that the CAF are the main source of CXCL12 β , to place this CXCL12 β /CXCR4 axis in the context of other Treg chemokine/receptor axes defined in ovarian cancer (e.g., CCR4/CCL17, CCL28/CCR10), and showing that cells attracted are suppressive and especially how they compare to Tregs attracted through the other defined axes. In that way, the contributions to immunosuppression will be convincing as suggested by the title of the manuscript.

Figure 1 evidence for 4 distinct FACS-identified CAF in HGSC is good. Evidence that mesenchymal HGSC accumulate CAF-S4 is good. What are the data for why this particular CAF accumulates in HGSC?

Figure 2 evidence that CAF are more abundant in stroma versus epithelium is good. Evidence that CAF-S1 preferentially accumulate Foxp3+ cells in HGSC stromal tissue using IHC over CAF-S4 is good. The evidence that these Foxp3+ cells are Tregs is not shown in this data set. The statement that "HGSC enriched in CAF-S1 fibroblasts are highly infiltrated by immunosuppressive regulatory T cells" is thus not supported, for the further reason that function of these cells is not shown.

Figure 3 Data that S1 and S4 CAF have distinct molecular signatures is good. Data support that message for CXCL12 β is increased over CXCL12 α is supportive although bigger numbers would make a stronger argument. Evidence that CXCL12 β message exceeds CXCL12 α message in HGSC is good. This statement "these data indicate that the CXCL12 β isoform is more expressed than CXCL12 α in mesenchymal HGSC, consistent with the accumulation of CAF-S1 fibroblasts..." is not fully supported as a role for other cells in producing CXCL12 β and recruiting Tregs is not reported here.

Reference to survival data based on CXCL12 α versus CXCL12 β isoforms is interesting. Please show

the survival by total CXCL12 in your data sets to confirm that total CXCL12 is not discriminatory and accords with prior total CXCL12 analyses.

Fig 4 Evidence that CAF-S1 can induce CD4+CD25+ T cell migration through CXCL12/CXCR4 is good. Evidence that these are Tregs by Foxp3 is shown, but no evidence that these are functional Tregs (e.g., suppressive) is shown. Further, the nature of non-migrating Tregs is not shown. Are non-migrating Tregs less functional or differ in other Treg functional markers (e.g., CTLA-4, Foxp3 MFI, CD25 MFI). How do these Tregs compare to those attracted through the other defined axes discussed above? What is the CXCL12 β contribution to immunosuppression versus the other Treg axes? The statement "CAF-S4 fibroblasts quickly died and could not be maintained in culture, thereby precluding any comparison between CAF-S1 and CAF-S4 fibroblasts in functional assays" refers to chemotaxis as the measured function. Thus, dead CAF-S4 cell proteins can still be used in Transwells to test a role for CAF-S1 in attracting Tregs. What is the level of silencing of CXCL12 α vs CXCL12 β and were multiple clones used, or a polyclonal culture? It would be useful to show if survival effect is contact-dependent using Transwells. The data that these Tregs are activated is not supported by any data provided.

Figure 5 Data that miR 141 and miR 200a regulate CXCL12 β is good. Data that miR200a is the active miR for CXCL12 β control in HGSOc is correlative but adequate.

Detailed reply to the reviewers' comments

Revised manuscript (NCOMMS-17-08140) Givel et al, "miR200-regulated CXCL12 β promotes fibroblast heterogeneity and immunosuppression in ovarian cancers"

We first would like to thank the Editorial Committee and the Receiving Editor for considering our work suitable for publication in *Nature Communications*, pending modifications. We were pleased to receive positive assessment of the three reviewers. We have greatly appreciated their comments that have contributed to improve the quality of our manuscript and enhance our message. We have now included in the new version of the text the modifications for addressing all their comments. For better visualization of the discussion, initial comments of the reviewers are below indicated in blue and our answers in Black.

Reviewers' comments:

Reviewer #1: (Remarks to the Author):

This is an ambitious and wide-ranging study which makes a number of novel and interesting claims. The major claims include identification of novel multi-marker-defined clinically relevant HGSOCA CAF subsets, of which one (CAF-S1) displays immune-suppressive actions, which at the molecular level are linked to an expression of CXCL12 β , which is suppressed in other CAF subsets through expression of miR200.

These are all interesting findings of biological and clinical potential. However, with reference to recent "houses of brick; mansions of straw" Kaelin comment in *Nature* (Vol. 545, page 387) authors and editors are encouraged to select a set of these claims for substantial validation in the preparation of a revised and more focused study.

Comments below are focusing on the first two of the claims identified above, which are estimated to be of most interest.

We are grateful to the reviewer for his/her positive evaluation of our work. We have now addressed the main points he/she underlined, which improved the quality of the manuscript. We are very pleased to list below the main modifications inserted into the text and figures.

Major points:

The novelty in the approach of a six-marker-based FACS characterization of clinically derived CAFs is recognized.

A. The subset classification implies the existence of specific marker combinations (e.g. S2 cells with FSPmed/high and low SMA, FAP and CD29) and the absence of cells with other possible combinations (e.g. "only-caveolin-high"). These findings should be validated by selected sets of double- or triple IF analyses of clinical samples.

We thank the Reviewer for this suggestion. As requested, we have now introduced, in the new version of the manuscript, immunofluorescence analyses using triple staining of high-grade serous ovarian cancer (HGSOCA) samples. As our study is mainly focused on the two myofibroblastic CAF subsets (CAF-S1 and CAF-S4) that accumulate the most in HGSOCA, we combined the staining of FAP, CD29 and SMA markers on the same human tumor sections. The combined triple IF enabled us to visualize CAF-S1 and CAF-S4 cells in HGSOCA, and thus validated our findings on clinical samples using triple IF analyses, as requested. These staining have now been included in the **new Figure 2G**, and described **p6**.

B. The CAF classification in 1D-G and K is done on an individual cell basis and defining features are summarized in 1E-F. However, the classification of cases in H-J appears to be derived from a score established based on the marker status determined on a case-basis, rather than a cell-basis. The study would be improved with regard to stringency if the "case-classification" were derived from analyses of cell-based maps as those shown in Fig. 1 K.

As requested, we have performed this analysis considering that the visualization at cell-level could be informative. However, we need here to underline that these images at high magnification represent

only a part of the total tumor, while the global content of CAF subsets, established by combining CAF markers histological scores, is defined by analyzing the totality of each section. Moreover, these cell-based image analyses are difficult to proceed and time-consuming, thus the number of images analyzed (old 1K, new 2E Left and new Sup 1B) is lower than the total number of HGSOC patients analyzed by histological scoring (old 1I, new 2C). All these technical information have now been specified in the new version of the text. Nevertheless, as recommended by the Referee, we have performed the cell-based analysis. Interestingly, and despite the technical limitations mentioned above, we found similar results in the content of CAF subsets using this method. We thus thank the reviewer for this suggestion and we have now included these results in the **new Figure 2E, Right**.

In addition, and to give a better view of our data, we have introduced several modifications in the new version of the text. We provide now a better description of the strategy used for creating the CAF maps at cellular scale in the **Method section, p 20, #Development of a decision tree algorithm for prediction of CAF subset identity**. Moreover, we have introduced several precisions in the description of the results (**new Figure 2B-E, p6-7**) and the corresponding Figure 2 **legend (p28-29)**. Among other information, we now underline the number of HGSOC samples (including prospective and retrospective cohorts) from which the data have been established, to avoid any misinterpretation in the number of samples included in our study. Finally, we also provide additional examples of CAF maps from different HGSOC samples (**New Supplemental Figure 1B**). These are important issues and we thank the reviewer for his/her comments, which provide substantial help in improving this part of the manuscript.

C. The S1 subset is mostly defined by its FAP-high status. This should be further high-lighted.

As requested, we have now provided this information all along the text. Moreover, as we studied several other CAF markers, in addition to FAP, we have also included the protein levels of the different markers in the 4 CAF subsets. We have now thus described in a more appropriate way the different CAF subsets, regarding the combination of the 6 CAF markers used in our study (**p6** and **p13**, for example and the corresponding Figure legends, **p28**).

In addition, we now give the list of genes specifically up-regulated in CAF-S1 fibroblasts, cells that were isolated fresh HGSOC samples and directly analyzed by RNAseq, thus precluding any artifact of cell culture. This CAF-S1-specific gene list provides new potential specific markers for the CAF-S1 subset. CAF-S1 gene signature is now introduced in the new version of the text (**New Supplemental Table 2, p8** and **p39**).

D. The mapping approach of Fig. 1K can be made very much more informative by adding information about spatial distribution of the CAF subsets with regard to vicinity to malignant cells, vasculature or immune cell clusters.

As requested, we have now provided quantitative data evaluating the distance between cancer cells and CAF-S1 fibroblasts. This quantification is based on an algorithm developed in the lab and implemented using R software. In brief, each tile of the images was annotated according to a specific position in the section, using Cartesian coordinate system in two dimensions (x and y coordinates). For each tile (annotated as a CAF subset based on CAF marker intensities), we calculated the distance with the closest epithelial cell (in a maximum area of 5 successive tiles in x and y) using Euclidean distance. We observed that there was a significant enrichment of CAF-S1 cells at close proximity of cancer cells, suggesting a mutual potential benefit between CAF-S1 fibroblasts and cancer cells, as recently shown in pancreatic cancers (Ohlund, et al., JEM 2017). These data have now been inserted in the **New Figure 2F** and described **p6** and **p29**, with precisions in the Method section (last part of **# Maps of CAF subsets, p20-21**).

As recommended by the Referee, we performed exactly the same type of analysis for defining the proportion of CAF subsets, regarding accumulation of CD3+ lymphocytes. These data are now shown in the **new Figure 3M** and described **p7**, with the corresponding Figure legend, **p30**.

Finally, we have also performed endothelial cells staining, using CD31 marker, in order to quantify blood vessel vasculature in CAF-S1- and CAF-S4-enriched HGSOC. We confirm that the density of CD31+ blood vessel is similar between these two types of HGSOC. These results are added in **Supplemental Figure 2B,C**, and described **p7**.

E. The identification of associations between the CAF-S1 subset and immune profile (Fig. 2) should acknowledge earlier literature which has implied immune-suppressive roles of FAP-positive fibroblasts. Authors should more explicitly define how the findings of the present study go beyond earlier literature on immune-suppressive effects of FAP+ CAFs, including refs 35, 48 and e.g. Yang, Lin et al., *Can Res*, 2016 (not cited).

As requested, we have now discussed in a more detailed way the results obtained from earlier literature and have cited the work mentioned by the Referee (**Discussion, p14**). In addition, we provided now a substantial number of additive data highlighting the role of CAF-S1 fibroblasts on immunosuppression.

First, we performed immunosuppressive assays and have tested the capacity of CD4+CD25+ (CD25^{High} CD127^{low} CD45RA^{low}) T-lymphocytes following co-culture with CAF-S1 fibroblasts to inhibit the proliferation of CD4+ effector T-cells. We established that the suppressive activity of these T-lymphocytes was significantly enhanced upon co-culture with CAF-S1 fibroblasts, suggesting that CAF-S1 primary fibroblasts (isolated from HGSOC patients) could promote immunosuppression. These data are now described in the **New Figure 5L** and described **p11**, with its corresponding legend **p32**.

Moreover, based on CAF-S1 RNAseq data, we identified candidate genes that could be involved in CAF-S1-mediated activity on T-lymphocytes. We have now tested these candidates and established the role of CD73, B7H3 and IL6 in CAF-S1-mediated function. Indeed, we show now that CAF-S1 fibroblasts enhance the survival, as well as the proportion of CD25+FOXP3+, through at least CD73, B7H3 and IL-6. These new data are now described in the new version of the manuscript (**New Figure 5M, N**, description **p11** and corresponding legend, **p32**).

F. The interesting S1/immune cell relationships discussed in Fig. 2 should be extended to analyses where spatial relationships between CAF and immune cell subsets are analysed; are there e.g. any evidence of restriction of a certain immune cell subsets in areas dominated by a particular CAF subset?

We have analyzed the proportion of CD3+ in HGSOC and found that CAF-S1 enriched tumors are highly infiltrated by CD3+ lymphocytes. As requested, we have completed this analysis by performing a spatial analysis of the distribution of CD3+ T-lymphocytes according to CAF subsets. To do so, by combining the analysis of CAF maps and the staining of CD3+ T-lymphocytes, we have now compared the number of CD3⁺ cells at the surface of each CAF subset within HGSOC sections. By this analysis at cellular level, we confirmed that the proportion of CD3⁺ T-lymphocytes was higher at the surface of CAF-S1-cells compared to any of the other CAF subsets. These analyses have now been included in the **new Figure 3M** and described **p7**, with its corresponding legend **p30**.

G. The data should allow analyses of possible correlations between outcome and abundance, or spatial distribution, of the CAF subsets. Such analyses appear more relevant and novel than the survival association reported for CXCL12beta (Fig. 3G).

As suggested by the Reviewer, we have analyzed the relationship between CAF subset accumulation and clinical outcome. As CAF-S1 and CAF-S4 accumulate the most in HGSOC, we tested if accumulation of one of these subsets could be linked to survival. However, we did not find any link. We obtained preliminary data suggesting that CAF-S4 are related to metastatic spread in human breast cancers, further suggesting that both CAF-S1 and CAF-S4 could be associated with a dismal prognosis through distinct mechanisms, by increasing metastatic spread for CAF-S4, and inducing immunosuppression for CAF-S1, as describe here. In agreement with this notion, CAF-S1 and CAF-S4 are the two most frequent subsets detected in the mesenchymal HGSOC subtype, the most deleterious HGSOC subtype associated with poor patient survival. This is now more clearly stated into the text (**new Figure 2D**), **p6**.

Based on his/her first positive opinion on our data, and considering the substantial amount of data we have now provided, we hope that the reviewer will agree about the interest of the description of the immunosuppressive functions of CAF-S1 fibroblasts in HGSOC.

Reviewer #2 (Remarks to the Author):

This is an original and carefully conducted study of fibroblasts in high grade serous ovarian cancer, HGSC that contributes to our understanding of the role of fibroblasts in the tumor microenvironment and also extends our knowledge of the role of CXCR4 and CXCL12 isoforms in this disease. It is commendable that all work has been conducted on primary cells isolated from patients or human peripheral blood T cells.

We are very grateful to the Reviewer for his/her positive and constructive assessment about our work. We have now answered to the requested points and are delighted to share our answers with the Referee.

1. Were all the biopsies obtained from the patients prior to treatment? In some cases patients with HGSOC receive neo-adjuvant chemotherapy before surgery and this may affect fibroblast activation.

The point addressed here by the reviewer about the impact of the chemotherapy on CAF subsets is of particular importance. We would like first to emphasize that all HGSOC samples from the retrospective cohort included in our study have been collected prior to any treatment. Indeed the 118 HGSOC samples analyzed in the retrospective cohort are completely devoid of any effect of chemotherapy.

As indicated by the Reviewer, in France, although the number of HGSOC patients receiving neoadjuvant chemotherapy before surgery is significantly increasing, there is still a substantial number of HGSOC patients who are immediately operated. In the prospective cohort, we collected all available tumor samples, independently of the treatment prior surgery, as patient information is not systematically accessible at time of surgery. Thus, as requested by the Reviewer, we have now analyzed retrospectively the samples of the prospective cohort and compared CAF contents at time of surgery, with or without neo-adjuvant chemotherapy. To our surprise, the intensities of 5 out of the 6 CAF markers analyzed show no difference, regardless patients received or not chemotherapy before surgery. CD29 was the only marker with a slight but significant reduction in its intensity following treatment. Still, CD29 intensity remains the highest detected among CAF markers, even after treatment and we do not detect any difference between CAF-S1 and CAF-S4 subsets, the two most abundant CAF subsets in HGSOC. As these precisions are important to better appreciate our findings, we now provide the proportion of CAF subsets separating the samples with and without neo-adjuvant treatment in **Supplemental Figure 2D** and give details of our procedure in the **Methods** section, end of paragraph *#Cohorts of HGSOC patients*, **p17**.

2. Can the authors justify the statistical methods used in some of the data analysis? For instance, they use of parametric tests on data that look non-normal, and T-tests for multiple comparisons.

We completely agree with the Reviewer and we apologize for the errors in the previous version of our manuscript. We have now tested again the normality of the data distribution and verified for all presented data that the test applied was adapted. Thus, statistical tests used are in agreement with data distribution: Normality was first checked using the Shapiro–Wilk test and parametric or non-parametric two-tailed test was applied according to normality respect. When normal distribution was observed, equality of variances was then tested using Bartlett's test. If variances between groups were similar, Student's t-test was performed. Differences were considered to be statistically significant at values of $P \leq 0.05$ (bilateral tests). Spearman's correlation test was used to evaluate the correlation coefficient between two parameters. As required, we have indicated the statistical tested used in each figure legend. Moreover, we dedicated one paragraph at the end of the **Method** section (*# Statistical analysis*, **p26**), to clarify the method used and the statistical tests applied all along the study.

3. As chemokines bind to ECM proteins and localising them in tissue is not always easy using IHC, the use of in situ hybridisation e.g. with RNAscope to detect CXCL12 mRNA would be a useful addition to this paper.

As requested, we have now investigated CXCL12 expression in human HGSOC sections by *in situ hybridization* using RNAscope[®] Technology, as suggested by the reviewer. Thanks to this method, we confirmed that CXCL12 is mainly expressed by stromal cells, as the positive staining for CXCL12 is strictly detected in fibroblasts. The technology is now described in the Method section *#CXCL12 in situ hybridization*, **p23**. Moreover, the data have been included in the **New Figure 4M** and described **p9**, with its corresponding legend, **p31**.

In addition, we analyze *CXCL12* expression levels among different cell types by analyzing publicly available dataset. In that aim, we used Gene Expression Omnibus (GEO) resources and expression data from Atlas of Human primary cells (GSE49910). We observed that *CXCL12* mRNA level is very high in fibroblasts and much less detected, or even at background level, in other cell types, such as immune cells, including lymphoid and myeloid cells. This suggests that *CXCL12* expression mainly results from fibroblasts. The table corresponding to this analysis has now been added in **Supplemental Figure 5** and described **p9**, data in total agreement with the expression pattern observed using RNAscope technology.

Reviewer #3 (Remarks to the Author):

This manuscript by Givel and colleagues assesses the role of CAF in HGSOc immunopathogenesis. The manuscript asserts that a specific CAF-S1 population attracts immunosuppressive Tregs through *CXCL12/CXCR4* and that this is regulated by miR200a in the CAF. Much interesting data is presented and the work is relevant and topical. However, there are loose mechanistic ends that reduce the potential impact of the work in its current form. The biggest issues are showing that the CAF are the main source of *CXCL12 β* , to place this *CXCL12 β /CXCR4* axis in the context of other Treg chemokine/receptor axes defined in ovarian cancer (e.g., *CCR4/CCL17*, *CCL28/CCR10*), and showing that cells attracted are suppressive and especially how they compare to Tregs attracted through the other defined axes. In that way, the contributions to immunosuppression will be convincing as suggested by the title of the manuscript.

We thank the reviewer for his/her positive evaluation of our work. We have greatly appreciated his/her concerns and have done our best to answer to his/her comments.

Figure 1 evidence for 4 distinct FACS-identified CAF in HGSOc is good. Evidence that mesenchymal HGSOc accumulate CAF-S4 is good. What are the data for why this particular CAF accumulates in HGSOc?

We thank the reviewer for his/her good assessment of this part of our work. We observe that the CAF-S1 subset significantly accumulates in mesenchymal HGSOc, not the CAF-S4 cells, thus we assume here that the Reviewer meant considering CAF-S1. The mechanism(s) driving CAF-S1 accumulation in mesenchymal HGSOc (and CAF-S4 in non-mesenchymal tumors) is an interesting, although challenging, question. Here, we have tried to address -at least in part- this point by highlighting a new role of the miR-200 family members in this particular subtype. We and others have previously demonstrated that miR-200 expression is increased in ovarian cancers. Moreover, expression of this family of miRNA has been one of the first mechanisms identified that allows the distinction between the mesenchymal *versus* non-mesenchymal molecular HGSOc subtypes. Moreover, it has been shown already that the expression of this miRNA family is dependent on oxidative stress in ovarian cancers. Here, we provide additive insights showing that this already-established regulation has also some impact on the surrounding stromal cells. Indeed, we show that the *CXCL12 β* isoform specifically accumulates in the CAF-S1 subpopulation, and not in the CAF-S4 subset. This differential accumulation results from a post-transcriptional mechanism, dependent of miR-200 family members, miR-141 and miR-200a. The expression of these two miRNA leads to the specific down-regulation of the *CXCL12 β* isoform in CAF-S4 fibroblasts and subsequently to its accumulation in CAF-S1 immunosuppressive fibroblasts. Although we provide here important new insights in deciphering -at least in part- the mechanisms driving to CAF-S1 accumulation in mesenchymal HGSOc: Lower ROS compared to non-mesenchymal - Reduced miR-200 expression - Increased *CXCL12 β* - stronger attraction of T-lymphocytes. Still, the reasons why there is a discrepancy in oxidative stress between mesenchymal and non-mesenchymal HGSOc are not known. As requested, we have tried to discuss these different notions into the **new version of the text**, in the result section **p12**, and in the discussion **p15**.

Figure 2 evidence that CAF are more abundant in stroma versus epithelium is good. Evidence that CAF-S1 preferentially accumulate Foxp3+ cells in HGSOc stromal tissue using IHC over CAF-S4 is good. The evidence that these Foxp3+ cells are Tregs is not shown in this data set. The statement that "HGSOc enriched in CAF-S1 fibroblasts are highly infiltrated by immunosuppressive regulatory T cells" is thus not supported, for the further reason that function of these cells is not shown.

Many thanks to the Reviewer for his/her positive assessment about the assumptions listed in the first part of his/her comment, as these data are based on patient samples that are always difficult to

collect, in particular in association with all clinical data. We do agree with the Reviewer concerning the second part of his/her comment. Although FOXP3+ T-cells could be enriched in regulatory T-cells, the single staining of FOXP3+ is not sufficient by itself, and we apologize for that shortcut. We have now amended the text appropriately, as requested by the Reviewer (description of **new Figure 3, p6-7**).

In order to address the impact of CAF-S1 on the immunosuppressive function of T cells, we have now performed immunosuppressive assays. Indeed, we have tested the capacity of CD4+CD25+ (CD25^{High} CD127^{low} CD45RA^{low}) T-lymphocytes following co-culture with CAF-S1 fibroblasts to inhibit the proliferation of CD4+ effector T-cells. We established that the suppressive activity of these T-lymphocytes was significantly enhanced upon co-culture with CAF-S1 fibroblasts, suggesting that CAF-S1 primary fibroblasts (isolated from HGSOC patients) could promote immunosuppression. These data are now described in the **New Figure 5L** and described **p11**, with its corresponding legend **p32**.

Figure 3 Data that S1 and S4 CAF have distinct molecular signatures is good. Data support that message for CXCL12 β is increased over CXCL12 α is supportive although bigger numbers would make a stronger argument. Evidence that CXCL12 β message exceeds CXCL12 α message in HGSOC is good. This statement “these data indicate that the CXCL12 β isoform is more expressed than CXCL12 α in mesenchymal HGSOC, consistent with the accumulation of CAF-S1 fibroblasts...” is not fully supported as a role for other cells in producing CXCL12 β and recruiting Tregs is not reported here.

To try to address this point, we first sought to analyze CXCL12 expression levels among different cell types by analyzing publicly available dataset. In that aim, we used Gene Expression Omnibus (GEO) resources and expression data from Atlas of Human primary cells (GSE49910). We observed that CXCL12 mRNA level is very high in fibroblasts and much less detected, or even at background level, in other cell types, such as immune cells, including lymphoid and myeloid cells. This suggests that CXCL12 expression mainly results from fibroblasts. The table corresponding to this analysis has now been added in **Supplemental Figure 5** and described **p9**.

Moreover, we investigated CXCL12 expression in human HGSOC sections by *in situ hybridization* using RNAscope[®] Technology. Although we have not succeeded to perform this experiment using probes recognizing specifically each isoform, we confirmed that CXCL12 is mainly expressed by stromal cells, as the positive staining for CXCL12 mRNA is strictly detected in fibroblasts. The technology is now described in the Method section #CXCL12 *in situ hybridization*, **p23**. Moreover, the data have been included in the **New Figure 4M** and described **p9**, with its corresponding legend, **p31**.

Reference to survival data based on CXCL12 α versus CXCL12 β isoforms is interesting. Please show the survival by total CXCL12 in your data sets to confirm that total CXCL12 is not discriminatory and accords with prior total CXCL12 analyses.

We have now included into the new version the impact of the expression of total CXCL12 on patient survival. To do so, we considered the two CXCL12 isoforms detected in HGSOC (namely, CXCL12 α and CXCL12 β) and calculated the mean of their expression. Interestingly, considering both CXCL12 α and CXCL12 β expression in HGSOC, we observed a significant impact in the 3 independent cohorts of patients analyzed here. As CXCL12 β is a reliable prognostic factor in these cohorts, while CXCL12 α is not, these observations suggest the prognostic value of total CXCL12 followed this one of CXCL12 β , arguing for the important role of the CXCL12 β isoform in HGSOC. As requested, these observations are in agreement with previously published data and have been added in **Supplemental Figure 3E, p8**.

Fig 4 Evidence that CAF-S1 can induce CD4+CD25+ T cell migration through CXCL12/CXCR4 is good. Evidence that these are Tregs by Foxp3 is shown, but no evidence that these are functional Tregs (e.g., suppressive) is shown. Further, the nature of non-migrating Tregs is not shown. Are non-migrating Tregs less functional or differ in other Treg functional markers (e.g., CTLA-4, Foxp3 MFI, CD25 MFI). How do these Tregs compare to those attracted through the other defined axes discussed above? What is the CXCL12 β contribution to immunosuppression versus the other Treg axes?

As recommended by the Reviewer, we have tested whether the CD4+CD25+ cells exhibit immunosuppressive activity that can be modulated by CAF-S1. To do so, we isolated CD4⁺ CD25^{High} CD127^{low} CD45RA^{low} T-lymphocytes, put them in co-culture with CAF-S1 fibroblasts and next evaluated their impact on CD4+ effector T-cells. We found that the pre-culture of CD25^{High} CD127^{low} CD45RA^{low} T-lymphocytes with CAF-S1 fibroblasts significantly enhanced their impact on effector T-

cells by strongly inhibiting their division capacity. Thus, the suppressive activity of CD4+CD25+ T-lymphocytes, evaluated on CD4+ effector T-cells, was significantly enhanced upon co-culture with CAF-S1 fibroblasts, suggesting that CAF-S1 primary fibroblasts (isolated from HGSOc patients) could promote immunosuppression. These data are thus now described in the **New Figure 5L** and described **p11**, with its corresponding legend **p32**.

As suggested by the Reviewer, we have checked the expression of other chemokines in the RNAseq data that we generated from CAF subset subpopulations isolated from HGSOc (we now provide the CAF-S1 signature in the **Supplemental Table 2**). CCL17 and CCL22 were not detected, and CCL28 almost undetectable in CAF-S1 cells. In contrast, CCL2 was highly expressed in CAF-S1 cells, at a rate comparable to CXCL12. We thus tested the silencing of CCL2 in CAF-S1 cells on their capacity to attract CD4+CD25+ T-lymphocytes but found no impact of CCL2. These observations have been added in the **new Figure 5G** and described **p10**.

Finally, based on CAF-S1 RNAseq data, we identified candidate genes that could be involved in CAF-S1-mediated activity on T-lymphocytes. We have now tested these candidates and established the role of CD73, B7H3 and IL6 in CAF-S1-mediated function. Indeed, we show now that CAF-S1 fibroblasts enhance the survival, as well as the proportion of CD25^{High}FOXP3^{High} T-lymphocytes through CD73, B7H3 and IL-6. These new data are now described in the new version of the manuscript (**New Figure 5M, N** and **Results p11** and **p31**).

The statement “CAF-S4 fibroblasts quickly died and could not be maintained in culture, thereby precluding any comparison between CAF-S1 and CAF-S4 fibroblasts in functional assays” refers to chemotaxis as the measured function. Thus, dead CAF-S4 cell proteins can still be used in Transwells to test a role for CAF-S1 in attracting Tregs.

Although CAF-S4 died quickly, we consider the possibility to use CAF-S4 supernatants in transwell experiments. Unfortunately, CAF-S4 supernatants couldn't be used as well. Indeed, when we sort fibroblasts from fresh tumors, we generally get around a thousand cells for each subset. We thus directly sort the cells into 96-well plates in a total volume of 100uL of medium. Thus, technical issues preclude the use of sorted CAF-S4 supernatants: (1) we need 200uL of medium in each lower chamber of a transwell assay, and we furthermore always do at least duplicates; (2) we sort the cells into an enriched medium to help them recover after the procedure and the one we use in the transwell assay is DMEM-1% FBS which is not adapted to sorted cells. We are really sorry for that and hope that the reviewer will agree that to test CAF-S4 is really challenging and that this does not reduce the interest of our data on the role of CAF-S1 on immunosuppression, in particular considering the new data we now provide.

What is the level of silencing of CXCL12 α vs CXCL12 β and were multiple clones used, or a polyclonal culture?

The efficiency of CXCL12 silencing in CAF-S1 cells is shown in **new Figure 5E (p10, with the corresponding legend p31)**, where we can observe a strong reduction of CXCL12 α and/or CXCL12 β mRNA levels, in agreement with the siRNA used. Several primary CAF-S1 cell lines were used in this paper, as these cells become senescent after few passages. In all cases, the efficiency of the silencing was equivalent to this one shown in the Figure. The information related to CAF-S1 culture is now indicated in the **Method** section # *Primary ovarian CAF culture*, **p21**.

It would be useful to show if survival effect is contact-dependent using Transwells. The data that these Tregs are activated is not supported by any data provided.

As asked by the reviewer, we have observed that the impact of CAF-S1 on survival of CD4+CD25+ T-Lymphocytes is indeed contact-dependent. As suggested, we performed Transwell assays and found that the CAF-S1-mediated pro-survival effect is not observed. We added this data in the **new Figure 5B**. Moreover, as mentioned before, we change the text appropriately to avoid confusion about T-reg activation.

Figure 5 Data that miR 141 and miR 200a regulate CXCL12 β is good. Data that miR200a is the active miR for CXCL12 β control in HGSOc is correlative but adequate.

Here again, we thank the reviewer for his/her positive assessment of our work and careful reading. We have done our best to convince him/her and hope he/she will agree that our data deserve publication.

REVIEWERS' COMMENTS:

Reviewer #1 (Remarks to the Author):

The revised version of this manuscript contains important additions and improvements.

Novel Figs 2E, 2F, 3M well responds to many of the previous concerns (earlier main points (B, D and F)

Some other issues have been addressed by text changes and other new analyses (C, E, G).

Major remaining concern is the novel Fig. 2G, related to earlier main point A. Layout or representation of data should be modified to make a more convincing case that the two marker-defined S1 (CD29med/hi; FAP hi; SMA med/hi) and S4 (CD29 hi; FAP low; SMA high) subsets are confirmed by the triple FAP/CD29/SMA IF analyses.

Reviewer #2 (Remarks to the Author):

The authors have answered all the reviewers questions and critiques in a satisfactory manner.

Reviewer #3 (Remarks to the Author):

Reviewed by Tyler Curiel, UT Health San Antonio, TX USA

In this revised manuscript, the authors have done an extremely commendable job in addressing all prior critiques. Explanations for updated data and the limitations of the technology and conclusions are also well described and thoughtful. The revised manuscript provides an interesting and compelling story with sufficient data to support all major claims well. These findings represent important advances in understanding the role of SAF in cancer immunopathogenesis and will be of interest to a broad readership.

The statistical analyses were re-done at reviewer request and are appropriate to support the conclusions.

This reviewer still requests some minor edits to discussion and interpretation as stated in the detailed comments.

Specific comments.

Figure 1. The mechanisms for CAF-S1 accumulation in HGSOC remains incompletely understood from the follow-up work, but the limitations of getting a better definition now are justified, and do not detract from then overall significance of the data. Thus, although incomplete, the lack of full mechanism at this stage is scientifically justified and acceptable.

Figure 2. The revised data here partially address the full identity of Foxp3+ cells and advance the understanding of CAF effects on them. It is still not clear that Tregs in situ are more suppressive in the presence of one versus another type of CAF, but the in vitro data support this concept. Again, although incomplete, the lack of full understanding of the functionality of in situ Tregs based on CAF effects at this stage is scientifically justified and acceptable. However, since in vivo differences in functionality have not been established, the interpretation/discussion should be modified to state that it is "likely" or "possible" that these Tregs in situ are more suppressive but not to assert that they definitely are.

In figure 3, the updated data on the source of CXCL12, survival effects of CXCL12 total and isoforms are sufficient, and very nicely done.

In figures 4-5, the Treg trafficking effects of CAF are well done and sufficient. The new data on CD73, IL-6 and B7-H3 effects on Treg function are an interesting complement to these and prior requested data on Treg functional effects. However, B7-H3 effects are quite modest. Although statistically significant, it is not clear that this is biologically relevant. CD73 and IL-6 effects deserve some discussion, especially as this reviewer is not aware that IL-6 has any direct effect on Treg differentiation.

The inability to distinguish CAF-S1 versus CAF-S4 effects on migration are justified, and lack of these data do not significantly diminish the overall story. Gene silencing data and methods used here, and Treg survival effects data are now well described and sufficient.

In reviewing responses to the other two referees' comments and critiques, the authors appear to have adequately addressed all of them with sufficient rigor or provided justified arguments why additional studies could not be fully performed.

Detailed reply to the reviewers' comments

Revised manuscript (NCOMMS-17-08140A) Givel et al, "miR200-regulated CXCL12 β promotes fibroblast heterogeneity and immunosuppression in ovarian cancers"

We first would like to thank the Editor and the three Reviewers for considering our work for publication in *Nature Communications*. We are pleased to receive their positive assessment. We are very grateful to the three Reviewers for their positive and constructive comments about our work. We have appreciated their comments that contributed to improve the quality of our manuscript.

Reviewers' comments:

Reviewer #1 (Remarks to the Author):

The revised version of this manuscript contains important additions and improvements.

Novel Figs 2E, 2F, 3M well responds to many of the previous concerns (earlier main points (B, D and F))

Some other issues have been addressed by text changes and other new analyses (C, E, G).

Major remaining concern is the novel Fig. 2G, related to earlier main point A. Layout or representation of data should be modified to make a more convincing case that the two marker-defined S1 (CD29^{med/hi}; FAP^{hi}; SMA^{med/hi}) and S4 (CD29^{hi}; FAP^{low}; SMA^{high}) subsets are confirmed by the triple FAP/CD29/SMA IF analyses.

We thank the reviewer for his/her positive assessment about the changes that we introduced in the revised version of the manuscript.

As requested, we have now replaced the layouts in the Fig. 2G. We have inserted images at higher magnification that provide much better visualization of CAF-S1 and CAF-S4-enriched high-grade ovarian cancers.

Reviewer #2 (Remarks to the Author):

The authors have answered all the reviewers questions and critiques in a satisfactory manner.

We are very grateful to the reviewer for his/her positive evaluation of our work.

Reviewer #3 (Remarks to the Author):

In this revised manuscript, the authors have done an extremely commendable job in addressing all prior critiques. Explanations for updated data and the limitations of the technology and conclusions are also well described and thoughtful. The revised manuscript provides an interesting and compelling story with sufficient data to support all major claims well. These findings represent important advances in understanding the role of CAF in cancer immunopathogenesis and will be of interest to a broad readership.

The statistical analyses were re-done at reviewer request and are appropriate to support the conclusions.

This reviewer still requests some minor edits to discussion and interpretation as stated in the detailed comments.

Specific comments.

Figure 1. The mechanisms for CAF-S1 accumulation in HGSOV remains incompletely understood from the follow-up work, but the limitations of getting a better definition now are justified, and do not detract from their overall significance of the data. Thus, although incomplete, the lack of full mechanism at this stage is scientifically justified and acceptable.

Figure 2. The revised data here partially address the full identity of Foxp3⁺ cells and advance the understanding of CAF effects on them. It is still not clear that Tregs in situ are more suppressive in the presence of one versus another type of CAF, but the in vitro data support this concept. Again,

although incomplete, the lack of full understanding of the functionality of in situ Tregs based on CAF effects at this stage is scientifically justified and acceptable. However, since in vivo differences in functionality have not been established, the interpretation/discussion should be modified to state that it is “likely” or “possible” that these Tregs in situ are more suppressive but not to assert that they definitely are.

In figure 3, the updated data on the source of CXCL12, survival effects of CXCL12 total and isoforms are sufficient, and very nicely done.

In figures 4-5, the Treg trafficking effects of CAF are well done and sufficient. The new data on CD73, IL-6 and B7-H3 effects on Treg function are an interesting complement to these and prior requested data on Treg functional effects. However, B7-H3 effects are quite modest. Although statistically significant, it is not clear that this is biologically relevant. CD73 and IL-6 effects deserve some discussion, especially as this reviewer is not aware that IL-6 has any direct effect on Treg differentiation.

The inability to distinguish CAF-S1 versus CAF-S4 effects on migration are justified, and lack of these data do not significantly diminish the overall story. Gene silencing data and methods used here, and Treg survival effects data are now well described and sufficient.

In reviewing responses to the other two referees’ comments and critiques, the authors appear to have adequately addressed all of them with sufficient rigor or provided justified arguments why additional studies could not be fully performed.

We thank the reviewer for having considered that we have appropriately addressed most of his concerns in the previous version of our manuscript. In addition, we have now answered to the remaining points by modifying the text, as requested. Indeed, we have mentioned all along the text the fact that CAF-S1 are only likely to exert immunosuppressive functions *in vivo*. Moreover, we have now mentioned the role of B7H3, CD73 and IL6 in the discussion, as recommended. We hope that the modifications will be satisfying and allow definitive agreement for publication of our manuscript.